# Physical distancing and emergency medical services utilization after self-harm in Korea during the early COVID-19 pandemic: A nationwide quantitative study

**Ye Ji Lee[1], Min A. Yuh[1], In Soo Kim[1], Byul Nym Hee Cho[1], Seon Hee Woo[2], Sungyoup Hong[1]\***

1 Department of Emergency Medicine, Daejeon St Mary's Hospital, The Catholic University of Korea College of Medicine, Seoul, Republic of Korea, 2 Department of Emergency Medicine, Incheon St Mary's Hospital, The Catholic University of Korea College of Medicine, Seoul, Republic of Korea

\* emhong@catholic.ac.kr

## Abstract

### Background

People experienced various stress and psychological responses to the coronavirus disease 2019 (COVID-19) pandemic. This study aimed to examine the changes in emergency medical services (EMSs) utilization by self-harm patients in early pandemic and the impacts of physical distancing measures on the EMSs utilization by self-harm patients.

### Methods

Data for all patients presenting to emergency departments (EDs) after self-harm injuries including self-poisoning were collected from the National ED Information System (NEDIS). Characteristics of patients in two study regions (urban versus rural) were compared. Weekly and annual ED visit rates after self-harm (VRSH) per 100,000 population were calculated. Mobile phone mobility index (MPMI) was calculated by dividing a region's aggregated mobile phone mobility by mid-year population. Joinpoint regression analysis was conducted to assess changes in 2020 over pre-pandemic years. Test for presence of joinpoint at the end of 2019 was performed. A cross-correlation function was used to estimate the maximal morphological similarity and lag time between changes in MPMI and VRSH.

### Results

In 2020, in early phases of the pandemic, there was a moderate decline in self-harm-related ED visits to 30,797 from a continuously increasing trend seen in previous years. However, proportions of young people (50.1%) and females (62.3%) increased over previous years. VRSHs among women and young people aged 15–34 years showed higher levels in 2020 than in previous five years. There was a significant decrease in the proportion of patients transported directly from the scene. In addition, there was a polarization of mental state upon ED arrival from alert and unresponsive. The median correlation coefficient between

available using the following link; https://doi.org/
10.5281/zenodo.1481223.

**Funding:** The authors received no specific funding
for this work.

**Competing interests:** The authors have declared
that no competing interests exist.

MPMI values and VRSH values was 0.601 (interquartile range [IQR]: 0.539–0.619) in urban
regions and 0.531 (IQR: 0.454–0.595) in rural regions, showing no statistically significant dif-
ference between the two.

## Conclusion

Physical distancing measures adopted to prevent the spread of transmittable diseases fol-
lowing the pandemic had the effect of decreasing ED visits due to self-harm. When the pan-
demic has ended, and daily life has been restored, it will be particularly important to pay
attention to the increased numbers of self-harm patients expected to visit EDs compared to
during the pandemic.

## Introduction

Since the first report of the emergence of the novel coronavirus disease 2019 (COVID-19) in
Wuhan, Hubei, China, in December 2019, 754 million cases of this disease have been con-
firmed worldwide [1]. The government of each country implemented physical distancing strat-
egies, such as stay-at-home orders, bans on large gatherings, and the closure of schools and
businesses, to control the spread of the virus [2]. According to previous studies, limiting the
size of home ranges was effective in decreasing the number of infected individuals [3]. In an
Eastern Asian study, physical distances were found to be effective in reducing disease trans-
mission, and in reducing mobility in urban regions more than in rural regions [4].

A variety of emotional distresses were reported in response to the COVID-19 pandemic.
According to a study conducted in China between January 31 and February 2, 2020, 16.5% of
the participants had moderate-to-severe depressive symptoms, and 28.8% had moderate-to-
severe anxiety symptoms as a result of COVID-19 [5]. The vast majority of people experienced
social isolation, hardships on a daily basis, and rumors leading to anxiety and the fear of being
trapped during a mass quarantine [6]. People infected with the virus may feel fear, self-blame,
or guilt as well as be subjected to discrimination from others [7]. The findings from polls con-
ducted during the first month of the pandemic in the United States (US) showed that over
40% of adults reported a negative impact of the pandemic on their mental health, with 19%
describing this impact as "major" [8]. COVID stress syndrome is a term used to describe severe
stress and anxiety symptoms caused by COVID-19 [9]. Identifying how mental health
resources should be allocated to combat psychological effects after COVID-19 requires the col-
lection of specific information regarding mental health complications. Studies in Korea
reported that young and elderly adults felt fatigued, stressed, and had psychiatric problems,
such as depression, related to the COVID-19 pandemic [10, 11].

The term "self-harm" refers to nonfatal intentional self-injury or self-poisoning, regardless
of apparent motivation or suicidal intent [12]. It is widely accepted that self-harm is one of the
strongest predictors of suicide [13]. Mental health systems emphasize interventions that pro-
mote social connections, and research found that peer connections conferred a level of protec-
tion against self-harm in a variety of conditions [14–16]. Social isolation and loneliness are
also known to contribute to suicidal thoughts and self-harm [17]. Therefore, preventing physi-
cal distancing consequences was one of the most critical public health strategies during the
COVID-19 outbreak [18]. However, the psychological effects of physical distancing have not
yet been fully analyzed.

This study aimed to examine characteristics of self-harm patients seen in emergency departments (EDs) during the COVID-19 pandemic and evaluate changes in emergency medical services (EMSs) utilization after self-harm compared to pre-pandemic. We also intended to determine whether the strength of physical distancing measures was related to utilization of EMS by self-harm patients. Thus, a cross-correlation analysis was conducted between EMSs utilization data collected from nationwide ED surveillance and mobile mobility data. Additionally, the present study examined the effect of physical distancing on EMS utilization by self-harm patients by using the ED visit rate after self-harm (VRSH) based on the characteristics of the region in which the patient resides divided into urban and rural regions.

## Methods

### Study design and data collection

In 2020, 401 EDs are open to the public without restrictions in 16 provinces across Korea. The Korean National Emergency Medical Center (NEMC) operates the Korean National ED Information System (NEDIS), which collects prospective data on all patients presenting to EDs. The NEDIS data set was gathered for all patients aged 15 and older who visited EDs for intentional self-injury or self-poisoning, regardless of the motivation or degree of suicide intent, and survived when they arrived at the ED from January 2015 until December 2020. Study data collected from the NEDIS database included age, sex, onset time, self-harm method, route taken to the ED, and outcome after emergency care. Patients with unknown injury onset, those who had been transferred from another hospital, and those who died in the ED were excluded from the study. All data were collected and stored in a secure environment without any collection of personal information.

Characteristics of the patients presenting to EDs after self-harm during the early pandemic period were compared by dividing them into rural and urban regions based on the location of the ED visited after self-harm. An urban region was defined as one with a population density of at least 1,000 inhabitants per square kilometer. A rural region was defined as one with a population density less than or equal to 1,000 inhabitants per square kilometer. We included Seoul, South Korea's capital, six metropolitan cities, and Gyeonggi province located near Seoul as urban regions. We also included eight other provinces as rural regions. By mid-2020 (1st July), 32,778,938 people over 14 years of age lived in urban regions and 14,736,844 people lived in rural regions.

Population data for each region (province), including annual mid-year population, age, and sex composition, were obtained from Statistics Korea (http://kostat.go.kr/portal/eng/index. action).

### Measurements

**Mobile phone mobility index (MPMI).** The MPMI was used as a proxy for physical distancing behaviors. The index refers to the migration of SK Telecom subscribers (24.1 million in January 2020), which is the largest mobile communications company in Korea and is based on the location of their communication base station. The subscriber's residence site was defined as the location of the base station that remained for the longest period from 0:00 to 06:00 during the survey period. One mobile mobility event was recorded if the subscriber moved out of the village (the smallest administrative unit in Korea) and remained there for more than 30 minutes. A return to one's residence site, however, was not considered a mobility event. The MPMI was calculated by aggregating the total mobile mobility for the population aged over 14 years in each province and dividing it by the mid-year population aged over 14 years. A set of MPMI data was also obtained from Statistics Korea.

**VRSH.**   We obtained data on all individuals over 14 years old who visited EDs and received treatment due to self-harm, intentional self-injury, or self-poisoning regardless of the motivation or degree of suicidal intent from the NEDIS database from January 2020, when COVID-19 began to be more widespread in Korea, through December 2020. The major outcome of this study was weekly VRSH per 100,000 derived by dividing weekly number of ED visits after self-harm by annual mid-year population in each region. Annual VRSH was also calculated by dividing the annual number of ED visits after self-harm in each region by the annual mid-year population.

## Statistical analysis

All general characteristics variables, including sex, age, self-harm methods, route to the ED, and mental state upon ED arrival were compared between two study regions with a Chi-square test for independence. A Fisher's exact test was conducted when more than 20% of cells had expected frequencies < 5. The difference in MPMI value by week was compared between two years (2019 versus 2020) and two regions (urban versus rural) using two-way repeated measures analysis of variance (ANOVA). Week was used as a within-subject factor, while year and region were used as between-subject factors.

A joinpoint regression analysis was carried out to evaluate trend change in the number of self-harm visits and the proportion of patients according to general characteristics in the pandemic period compared to average annual trends for 2015–2019 [19]. In joinpoint regression, straight regression segments were joined at the "joinpoint" to identify changes in data trends. We set the minimum number of observations from a joinpoint at either end of the data to "1" and the minimum number of observations between joinpoints to "2" to locate the joinpoint at 2019. Average annual percentage change (AAPC) for 2015–2019 and annual percentage change (APC) for 2019–2020 as well as their 95% confidence intervals (CIs) were computed. It was determined whether a joinpoint existed at the end of 2019 by varying the number of joinpoints from 0 (minimum) to 1 (maximum). In this study, the null hypothesis was "0 joinpoint" and the alternative hypothesis was "1 joinpoint". T-statistics were calculated for the hypothesis that there was no change of slope between AAPC and APC at the joinpoint. If the p-value was less than 0.050, the null hypothesis was rejected and the hypothesis that there was a significant change in slope at the end of 2019 was accepted. The joinpoint graph was plotted over consecutive years to illustrate the split of data into two segments and its joinpoint (S1 Fig).

A cross-correlation function was performed to determine the degree of association between two variables (MPMI and VRSH) during the pandemic period. The cross-correlation function in R (https://rdrr.io/r/stats/acf.html) could measure the similarity between two time series as a function of their displacement from one another. The correlation coefficient (CC) between VRSH and MPMI was calculated at the point with the highest morphological similarity. The lag time was identified as the time gap (weeks) between two time series (S2 Fig). The sample size of CC values was very small in each group (eight provinces per group) to assume a normal distribution of data. The Shapiro-Wilk test was used to determine whether CC values were normally distributed (p > 0.050). Student t-tests were used if data satisfied a normal distribution. Otherwise, Mann-Whitney U tests were used.

Joinpoint regression analysis was conducted using a trend analysis software, Joinpoint version 4.9.1.0, developed by the National Cancer Institute (Information Management Services, Inc., Calverton, MD). All other statistical analyses were conducted using R 4.2.2 (The R Foundation for Statistical Computing, Vienna, Austria).

### Ethical approval

The Institutional Review Board of Daejeon St Mary's Hospital, The Catholic University of Korea reviewed and approved the study protocol (No DC22ZIS10009). The requirement for written informed consent was waived as the study was retrospective and the analyses used anonymous data from the NEDIS database that did not contain personally identifiable information.

## Results

Our analytic data set contained the nationwide emergency registry data of patients who visited ED for emergency care after self-harm. During the early pandemic period, 22,421 people visited EDs after self-harm in urban regions and 8,376 people visited EDs in rural regions (Table 1). Among them, 19,172 (62.3%) were females. The proportion of male patients was significantly higher in rural regions than in urban regions (p < 0.001). In the chi-square test for age-specific distribution showing significant association with study regions, age groups of 15–24 years and 25–34 years were overrepresented in urban regions, while self-harm patients aged 45 years and over were overrepresented in rural regions (p < 0.001). It was found that poisoning was the most common method of self-harm in both study regions and that it was overrepresented in rural regions than urban regions. Laceration/stabbing, hanging/chocking, struck by object, and fall were more overrepresented in urban regions, while submersion, burn, and traffic accident were more prevalent in rural regions (p < 0.001). The proportion of patients who visited EDs through outpatient units was higher in rural regions (p < 0.001). There was significant association between mentality upon ED arrival and the region in which EDs were located (p = 0.044). The proportion of patients in an alert mental state who attended EDs was higher than in urban regions, whereas rural regions had a higher proportion of unresponsive patients.

The weekly MPMI value was higher (+2.35%) in January 2020 than in January 2019. It decreased sharply at the first week of February (week 6 of 2020) and remained significantly lower than that in 2019 until it returned to the previous year's level in the first week of April (Fig 1). As the second wave of the pandemic began in mid-August (33rd week) in Korea, schools and workplaces were locked down, causing MPMI to drop sharply. As the nationwide pandemic occurred in mid-November (47th week), a continuous decrease in MPMI values was seen due to the lockdown until the end of the year. The MPMI for 2020 was significantly lower than the that for 2019 based on repeated measures ANOVA (degree of freedom (DF) = 1, F = 447.4, p < 0.001). MPMI in rural regions was statistically significantly lower than that in urban regions (DF = 1, F = 384.1, p < 0.001).

### ED self-harm visits during the early stages of the pandemic

Table 2 summarizes the characteristics of patients who self-harmed and their proportions based on variables. Self-harm-related ED visits increased from 23,254 in 2015 to 32745 in 2019 (average 1.5% increase for each year). However, visits decreased slightly to 30,797 in 2020 during the early pandemic period. As reported by Statistics Korea during the same period, suicides (the suicide rate per 100,000 people) increased to 13,092 (25.6), 13,463 (25.8), 13,670 (26.6), and 13,799 (26.9) in 2016, 2017, 2018, and 2019, respectively before declining to 13,195 (25.7) in 2020 [20].

The proportion of self-harm patients in urban regions significantly increased in 2020 compared to a declining trend in the previous five years according to joinpoint regression (Table 2, APC:1.8 [95% CI: 0.8, 2.5], p = 0.018). The proportion of female patients increased significantly in 2020 in contrast to previous years' trend (APC: 8.9 [95% CI: 3.5, 14.2], p = 0.014),

**Table 1. General characteristics of self-harm patients who visited emergency departments in the early pandemic (2020).**

| Variable | | Urban (%) | Rural (%) | Total (%) | Chi-square (P-value) |
|---|---|---|---|---|---|
| Sex (%) | Female | 14435 (64.6) | 4737 (56.6) | 19172 (62.3) | 158.7 (< 0.001) |
| | Male | 7986 (35.6) | 3639 (43.4) | 11625 (37.7) | |
| Age | 15–24 | 7006 (31.2) | 2154 (25.7) | 9160 (29.7) | 255.8 (< 0.001) |
| | 25–34 | 4778 (21.3) | 1473 (17.6) | 6251 (20.3) | |
| | 35–44 | 3394 (15.1) | 1343 (16.0) | 4737 (15.4) | |
| | 45–54 | 3194 (14.2) | 1370 (16.4) | 4564 (14.8) | |
| | 55–64 | 2014 (9.0) | 968 (11.6) | 2982 (9.7) | |
| | 65–74 | 939 (4.2) | 458 (5.5) | 1397 (4.5) | |
| | 75–84 | 869 (3.9) | 465 (5.6) | 1334 (4.3) | |
| | 85–94 | 216 (1.0) | 140 (1.7) | 356 (1.2) | |
| | 95- | 11 (0.0) | 5 (0.1) | 16 (0.1) | |
| Self-harm method | Poisoning | 12546 (56.0) | 5015 (59.9) | 17561 (57.0) | 65.2 (< 0.001) |
| | Laceration/ stabbing | 6815 (30.4) | 2268 (27.1) | 9083 (29.5) | |
| | Hanging/chocking | 893 (4.0) | 304 (3.6) | 1102 (3.6) | |
| | Struck | 822 (3.7) | 280 (3.3) | 1197 (3.9) | |
| | Fall | 401 (1.8) | 133 (1.6) | 534 (1.7) | |
| | Submersion | 207 (0.9) | 93 (1.1) | 300 (1.0) | |
| | Burn | 28 (0.1) | 15 (0.2) | 43 (0.1) | |
| | TA | 35 (0.2) | 31 (0.4) | 66 (0.2) | |
| | Other | 589 (2.6) | 218 (2.6) | 807 (2.6) | |
| | Unknown | 84 (0.4) | 19 (0.2) | 103 (0.3) | |
| | Machine | 1 (0.0) | 0 (0.0) | 1 (0.0) | |
| Route to ED | Direct | 22382 (99.8) | 8333 (99.5) | 30715 (99.7) | 26.5 (< 0.001) |
| | Outpatient | 37 (0.2) | 41 (0.5) | 80 (0.3) | |
| | Unknown | 2 (0.0) | 2 (0.0) | 4 (0.0) | |
| Mentality upon ED arrival | Alert | 15855 (70.7) | 5878 (70.2) | 21733 (70.6) | 8.0 (0.046) |
| | Verbal responsive | 3883 (17.3) | 1539 (18.4) | 5422 (17.6) | |
| | Pain responsive | 2269 (10.1) | 790 (9.4) | 3059 (9.9) | |
| | Unresponsive | 414 (1.8) | 169 (2.0) | 583 (1.9) | |
| Total | | 22421 (100.0) | 8376 (100.0) | 30797 (100.0) | |

ED, emergency department; TA, traffic accident.

while the proportion of male patients decreased from a gradual increase trend (APC: 1.4 [95% CI: 0.7, 2.1], p = 0.014) in the past. The proportion of self-harm patients in group aged 25–34 years increased significantly in 2020 (APC: 5.2 [95% CI: -10.7, 23.4], p < 0.001) but decreased significantly in age groups of 35–44 years, 45–54 years, 55–64 years, 65–74 years, 75–84 years, 85–94 years, and over 95 years (p = 0.047, p = 0.038, p = 0.041, p = 0.045, p = 0.015, p = 0.016, and p < 0.001, respectively). According to the self-harm method, only the proportion of laceration/stabbing injuries increased to 29.5% (APC: 8.8 [95% CI: -0.1, 20.0), p < 0.001), while methods such as hanging/choking and fall decreased significantly (APC: -23.0 [95% CI: -45.5, 8.8], p = 0.017 and APC: -23.0 [95% CI: -45.5, 8.8], p = 0.025, respectively). During the pandemic era, 96.3% of patients visited the ED directly, which was lower than the sustained trend of previous years (APC: -3.4 [95% CI: -3.5, -3.4], p = 0.001). In contrast, the number of patients brought to EDs from outpatient clinics increased significantly by 30-fold, from 0.1–0.2% to 3.4% (APC: 1643.4 [95% CI: 1077.0, 2324.0], p = 0.041). Compared with previous years, the

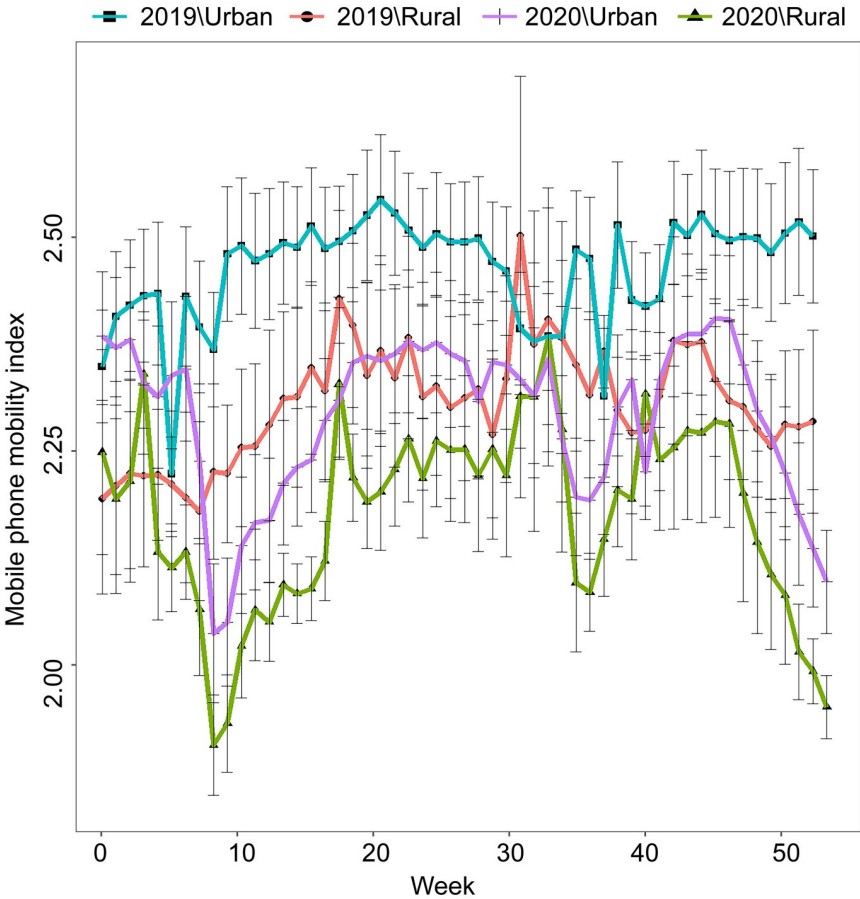

**Fig 1. Mobile phone mobility index (MPMI) values by week in Korea in 2020.** Points represent the arithmetic MPMI means for each province for each week and error bars indicate 95% confidence intervals. MPMI in rural regions was significantly lower than in urban regions (p < 0.001). MPMI dropped sharply during late February and August of 2020, when the community spread rapidly, and was sustained at a lower level in December following the nationwide spread (p < 0.001).

mental status of self-harm patients upon arrival at the ED showed a polarized pattern, showing significantly higher percentages of alert and unresponsive patients (APC: 6.4 [95% CI: 3.7, 8.3], p = 0.027 and APC: 54.7 [95% CI: 53.8, 55.6], p < 0.001, respectively). There were significant decreases in the percentage of patients responsive to verbal and pain stimulation in 2020 (APC: -39.8 [95% CI: -46.7, -14.5], p = 0.015 and APC: -45.9 [95% CI: -57.4, -31.3], p < 0.001, respectively).

Table 3 summarizes results of the trend analysis with joinpoint regression for population standardized VRSH. VRSH for males was 49.1 ± 21.5 per 100,000 people in 2020, demonstrating a reduction in contrast to significant increase trends over the previous five years (APC: -5.0 [95% CI: -9.5, -2.2], p = 0.042), whereas females' VRSH showed an increase in accordance with previous incremental trends (APC: 13.8 [95% CI: 8.2, 21.6], p = 0.242) without statistical significance. Standardized VRSH in 2020 were significantly increased in age group of 15–24 years (APC: 34.8 [95% CI: 14.7, 58.6], p = 0.049) compared to those of previous five years, but were significantly decreased in age groups of 45–54 years (APC: -9.4 [95% CI: -20.6, 2.8], p < 0.001), 75–84 years (APC: -15.4, p = 0.029), 85–94 years (APC: -15.4 [95% CI:-33.7, -2.1], p = 0.049), and 95 years and older (APC: -30.6 [95% CI: -41.8, -15.6], p = 0.030), respectively.

**Table 2. Comparison of the characteristics of self-harm patients in the early pandemic (2020) and previous years.**

| Variable | | 2015 (%) | 2016 (%) | 2017 (%) | 2018 (%) | 2019 (%) | 2020 (%) | Total (%) | AAPC 2015–19 (95% CI) | APC 2019–20 (95% CI) | P-value |
|---|---|---|---|---|---|---|---|---|---|---|---|
| Region | Urban | 16785 (72.2) | 16906 (71.5) | 17653 (71.2) | 20583 (70.5) | 23027 (70.3) | 22005 (71.5) | 114029 (71.2) | -0.7 (-0.9, -0.5) | 1.8 (0.8, 2.5) | 0.018 |
| | Rural | 6469 (27.8) | 6731 (28.5) | 7154 (28.8) | 8834 (29.5) | 9718 (29.7) | 8792 (28.5) | 46018 (28.8) | 1.7 (1.2, 2.3) | -4.3 (-6.8, -1.2) | 0.024 |
| Sex | Female | 12595 (54.2) | 12905 (54.6) | 13807 (55.7) | 16497 (56.1) | 18789 (57.3) | 19172 (62.3) | 93765 (56.9) | 1.4 (0.7, 2.1) | 8.9 (3.5, 14.2) | 0.014 |
| | Male | 10659 (45.8) | 10732 (45.4) | 11000 (44.3) | 12920 (43.9) | 13956 (42.6) | 11625 (37.7) | 70892 (43.1) | 5.4 (1.1, 10.0) | -1.8 (-2.7, -0.8) | 0.014 |
| Age | 15–24 | 4405 (18.9) | 4591 (19.4) | 5307 (21.4) | 7216 (24.5) | 7275 (22.2) | 9165 (29.8) | 37959 (23.1) | 8.3 (3.6, 13.2) | 7.2 (-20.4, 44.4) | 0.550 |
| | 25–34 | 4685 (20.0) | 4560 (19.3) | 4784 (19.3) | 5062 (17.2) | 6157 (18.8) | 6245 (20.3) | 31729 (19.3) | -0.5 (-3.8, 3.0) | 5.2 (-10.7, 23.4) | < 0.001 |
| | 35–44 | 4460 (19.2) | 4607 (19.5) | 4500 (18.1) | 5165 (17.6) | 5825 (17.8) | 4734 (15.4) | 29055 (17.6) | -2.5 (-7.1, 8.7) | -12.1 (-21.9, -1.9) | 0.047 |
| | 45–54 | 4271 (18.4) | 4364 (18.5) | 4564 (18.4) | 5205 (17.7) | 5249 (16.0) | 4564 (14.8) | 28217 (17.1) | -1.3 (-3.4, 2.6) | -9.1 (-14.7, -6.0) | 0.038 |
| | 55–64 | 2506 (10.8) | 2612 (11.2) | 2652 (10.7) | 3307 (11.2) | 3651 (11.1) | 2982 (9.7) | 17710 (10.8) | 0.9 (-0.5, 2.8) | -13.3 (-18.3, -5.9) | 0.041 |
| | 65–74 | 1467 (6.3) | 1417 (6.0) | 1403 (5.7) | 1534 (5.2) | 2129 (6.5) | 1400 (4.5) | 9350 (5.7) | -0.8 (-18.2, 36.7) | -21.9 (-43.5, 3.2) | 0.045 |
| | 75–84 | 719 (3.1) | 780 (3.3) | 863 (3.5) | 1024 (3.5) | 1615 (4.9) | 1334 (4.3) | 6335 (3.8) | -3.8 (-7.0, -0.4) | -56.4 (-62.0, -49.6) | 0.015 |
| | 85–94 | 719 (3.1) | 685 (2.9) | 718 (2.9) | 880 (3.0) | 822 (2.5) | 357 (1.2) | 4181 (2.5) | -3.8 (-7.0, -0.5) | -36.0 (-45.5, -22.4) | 0.016 |
| | 95– | 22 (0.1) | 21 (0.1) | 16 (0.1) | 24 (0.1) | 22 (0.1) | 16 (0.1) | 121 (0.1) | -7.4 (-27.9, 31.9) | -22.8 (-45.8, 2.0) | < 0.001 |
| Self-harm method | Poisoning | 13507 (58.1) | 13442 (56.9) | 14790 (59.6) | 15970 (54.3) | 19200 (58.6) | 17556 (57.0) | 94465 (57.4) | -6.6 (-50.6, 77.1) | 136.4 (-89.8, 319.8) | 0.552 |
| | Laceration/stabbing | 4906 (21.1) | 5379 (22.8) | 5396 (21.8) | 7788 (26.5) | 8453 (25.8) | 9087 (29.5) | 41009 (24.9) | 3.1 (-17.8, 34.4) | 8.8 (-0.1, 20.0) | < 0.001 |
| | Struck | 1101 (4.7) | 1172 (5.0) | 1050 (4.2) | 1271 (4.3) | 1295 (4.0) | 1139 (3.7) | 7028 (4.3) | -5.3 (-9.1, -1.3) | -6.9 (-7.2, 3.7) | 0.622 |
| | Hanging/chocking | 1720 (7.4) | 1683 (7.1) | 1622 (6.5) | 2058 (7.0) | 2245 (6.9) | 1193 (3.9) | 10521 (6.4) | 1.1 (-0.5, -1.4) | -23.0 (-45.5, 8.8) | 0.017 |
| | Fall | 626 (2.7) | 623 (2.6) | 576 (2.3) | 749 (2.5) | 640 (2.0) | 502 (1.6) | 3716 (2.3) | -9.0 (-14.0, -4.0) | -18.3 (-22.9, 12.5) | 0.025 |
| | Submersion | 302 (1.3) | 290 (1.2) | 282 (1.1) | 333 (1.1) | 336 (1.0) | 301 (1.0) | 1844 (1.1) | -5.5 (-6.3, -4.6) | -6.3 (-18.8, 8.0) | 0.942 |
| | Burn | 80 (0.4) | 76 (0.3) | 61 (0.2) | 73 (0.2) | 75 (0.2) | 43 (0.1) | 408 (0.2) | -11.8 (-18.9, -4.2) | -9 (-52.6, 74.4) | 0.390 |
| | TA | 97 (0.4) | 76 (0.3) | 68 (0.3) | 85 (0.3) | 74 (0.2) | 65 (0.2) | 465 (0.3) | -11.8 (-18.9, -4.2) | -11.8 (-18.9, -4.2) | 0.611 |
| | Machine | 21 (0.1) | 6 (0.0) | 7 (0.0) | 8 (0.0) | 9 (0.0) | 2 (0.0) | 53 (0.0) | -30.9 (-56.3, 9.2) | -9 (-15.4, -0.4) | 0.824 |
| | Other | 768 (3.3) | 732 (3.1) | 847 (3.4) | 867 (2.9) | 328 (1.0) | 806 (2.6) | 4348 (2.6) | -12.6 (-30.6, 9.9) | -28.8 (-99.3, 7619.6) | 0.725 |
| | Unknown | 126 (0.5) | 158 (0.7) | 108 (0.4) | 215 (0.7) | 90 (0.3) | 103 (0.3) | 800 (0.5) | -12.2 (-30.3, 10.2) | -17.0 (-100, 147233) | 0.773 |
| Route to ED | Direct | 23202 (99.8) | 23581 (99.8) | 24759 (99.8) | 29345 (99.8) | 32660 (99.7) | 29672 (96.3) | 163219 (99.1) | 0.0 (-0.0, 0.0) | -3.4 (-3.5, -3.4) | 0.001 |
| | Outpatient | 34 (0.1) | 39 (0.2) | 33 (0.1) | 58 (0.2) | 65 (0.1) | 1043 (3.4) | 1272 (0.8) | 8.2 (-1.3, 18.7) | 1643.4 (1077.0, 2324.0) | 0.007 |
| | Other | 17 (0.1) | 12 (0.1) | 12 (0.0) | 12 (0.0) | 15 (0.0) | 80 (0.3) | 148 (0.1) | -10.9 (-27.1, 3.6) | 545.1 (198.6, 994.0) | 0.014 |
| | Unknown | 1 (0.0) | 5 (0.0) | 3 (0.0) | 2 (0.0) | 5 (0.0) | 2 (0.0) | 18 (0.0) | 1.5 (-40.7, 73.5) | 1.5 (-40.7, 73.5) | 0.200 |

(*Continued*)

**Table 2.** (Continued)

| Variable | | 2015 (%) | 2016 (%) | 2017 (%) | 2018 (%) | 2019 (%) | 2020 (%) | Total (%) | AAPC 2015–19 (95% CI) | APC 2019–20 (95% CI) | P-value |
|---|---|---|---|---|---|---|---|---|---|---|---|
| Mentality upon ED arrival | Alert | 15248 (65.2) | 15611 (65.4) | 16529 (66.6) | 19443 (66.1) | 21567 (65.9) | 21681 (70.4) | 104148 (66.9) | 0.1 (-0.6, 0.6) | 6.4 (3.7, 8.3) | 0.027 |
| | Verbal responsive | 3802 (16.4) | 3903 (16.7) | 4188 (16.9) | 5109 (17.4) | 5304 (16.2) | 3110 (10.1) | 23844 (15.3) | 0.3 (-0.9, 1.6) | -39.8 (-46.7, -14.5) | 0.015 |
| | Pain responsive | 2391 (10.8) | 2421 (10.7) | 2436 (9.8) | 2852 (9.7) | 3029 (9.3) | 678 (2.2) | 12957 (8.3) | -2.6 (-4.1, -1.2) | -45.9 (-57.4, -31.3) | < 0.001 |
| | Unresponsive | 1808 (7.8) | 1700 (7.1) | 1652 (6.7) | 2013 (6.8) | 2816 (8.6) | 5328 (17.4) | 14755 (9.5) | -76.4 (-78.0, -75.2) | 54.7 (53.8, 55.6) | < 0.001 |
| | Unknown | 5 (0.0) | 2 (0.0) | 2 (0.0) | 0 (0.0) | 29 (0.1) | 0 (0.0) | 38 (0.0) | 1.5 (-35.0, 58.5) | 1.5 (-35.0, 58.5) | 0.198 |
| Total | | 23254 | 23637 | 24807 | 29417 | 32745 | 30797 | 164657 | | | |

Percentage represents the proportion of patients relative to the total number in each year. ED, emergency department; TA, traffic accident., AAPC, average annual percent change, APC, annual percent change, CI, confidence interval.

## Association between self-harm and physical distancing

In this study, we compared MPMI values, a quantitative indicator of physical distancing, and the VRSH per 100,000 people to estimate the psychological impact of the pandemic and physical distancing. We used a cross-correlation function to determine the relationship between MPMI values and VRSH.

Figs 2 and 3 illustrate cross-correlation between MPMI values and VRSH in Seoul and Busan (urban regions), respectively, and Jeonbuk and Gangwon (rural regions), respectively. The CC value was 0.601 (median, interquartile range (IQR); 0.539–0.619) in urban regions and 0.531 (IQR; 0.454–0.595) in rural regions, indicating strong correlations, but there was no significant difference in CC values between the two regions (Table 4, p = 0.092). There was no significant difference between urban and rural regions in terms of the CCs obtained based on male and female subpopulations (p = 0.156 and p = 0.495, respectively). In rural regions, the lag time between decreases in the MPMI and VRSH were 0.000 (IQR: 0.000–0.250) weeks,

**Table 3. Population standardized emergency department visit rate after self-harm (per 100,000 mid-year population) by age and sex.**

| Variable | | 2015 | 2016 | 2017 | 2018 | 2019 | 2020 | AAPC 2015–19 (95% CI) | APC 2019–20 (95% CI) | P-value |
|---|---|---|---|---|---|---|---|---|---|---|
| Sex | Female | 49.6 ± 14.7 | 50.6 ± 17.5 | 53.9 ± 20.6 | 64.3 ± 21.2 | 73.1 ± 23.8 | 78.4 ± 27.7 | 10.5 (7.2, 13.9) | 13.8 (8.2, 21.6) | 0.242 |
| | Male | 42.0 ± 12.2 | 42.1 ± 14.4 | 43.0 ± 22.1 | 50.5 ± 15.0 | 54.5 ± 23.2 | 49.1 ± 21.5 | 5.4 (1.1, 10.0) | -5.0 (-9.5, -2.2) | 0.042 |
| Age | 15–24 | 64.8 ± 28.6 | 67.5 ± 26.9 | 78.9 ± 35.6 | 110.4 ± 43.8 | 114.3 ± 48.9 | 158.6 ± 61.1 | 17.2 (-4.3, 48.5) | 34.8 (14.7, 58.6) | 0.049 |
| | 25–34 | 65.7 ± 25.2 | 69.2 ± 25.5 | 70.7 ± 30.8 | 76.2 ± 30.8 | 93.6 ± 31.4 | 99.1 ± 43.0 | 3.4 (-8.0, 10.3) | 12.6 (8.0, 25.2) | 0.114 |
| | 35–44 | 53.2 ± 22.6 | 52.7 ± 22.3 | 54.9 ± 23.4 | 63.7 ± 25.2 | 72.8 ± 24.9 | 64.0 ± 25.2 | 9.2 (-1.7, 44.4) | -9.0 (-28.2, 11.8) | 0.293 |
| | 45–54 | 49.5 ± 17.3 | 50.5 ± 19.2 | 52.8 ± 20.1 | 60.2 ± 21.5 | 60.4 ± 20.1 | 55.6 ± 23.2 | 6.1 (3.9, 19.3) | -9.4 (-20.6, 2.8) | <0.001 |
| | 55–64 | 40.4 ± 12.7 | 39.5 ± 10.5 | 37.6 ± 14.1 | 44.6 ± 12.2 | 47.7 ± 14.4 | 40.8 ± 20.6 | 3.0 (-4.2, 11.1) | 2.2 (-3.3, 8.2) | 0.530 |
| | 65–74 | 38.7 ± 11.9 | 36.4 ± 13.0 | 35.5 ± 14.7 | 38.0 ± 11.0 | 51.1 ± 8.5 | 35.9 ± 15.0 | 8.7 (-41.5, 130.6) | -25.9 (-75.5, 43.2) | 0.596 |
| | 75–84 | 35.5 ± 23.5 | 36.3 ± 20.6 | 37.9 ± 22.3 | 42.0 ± 20.9 | 62.7 ± 20.1 | 53.0 ± 19.0 | 12.1 (0.3, 25.3) | -15.4 (-33.7, -2.1) | 0.029 |
| | 85–94 | 159.5 ± 25.5 | 140 ± 29.1 | 136.6 ± 26.6 | 133.9 ± 23.8 | 131.3 ± 16.7 | 115.1 ± 22.1 | -4.3 (-11.0, 3.0) | -11.0 (-22.2, 7.3) | 0.049 |
| | 95- | 83.6 ± 48.1 | 72.8 ± 45.8 | 50.1 ± 43.6 | 66.3 ± 50.6 | 53.7 ± 26.0 | 38.3 ± 15.8 | -9.9 (-12.9, 0.5) | -30.6 (-41.8, -15.6) | 0.030 |

AAPC, average annual percent change; APC, annual percent change; CI, confidence interval. Data are presented as mean ± SD.

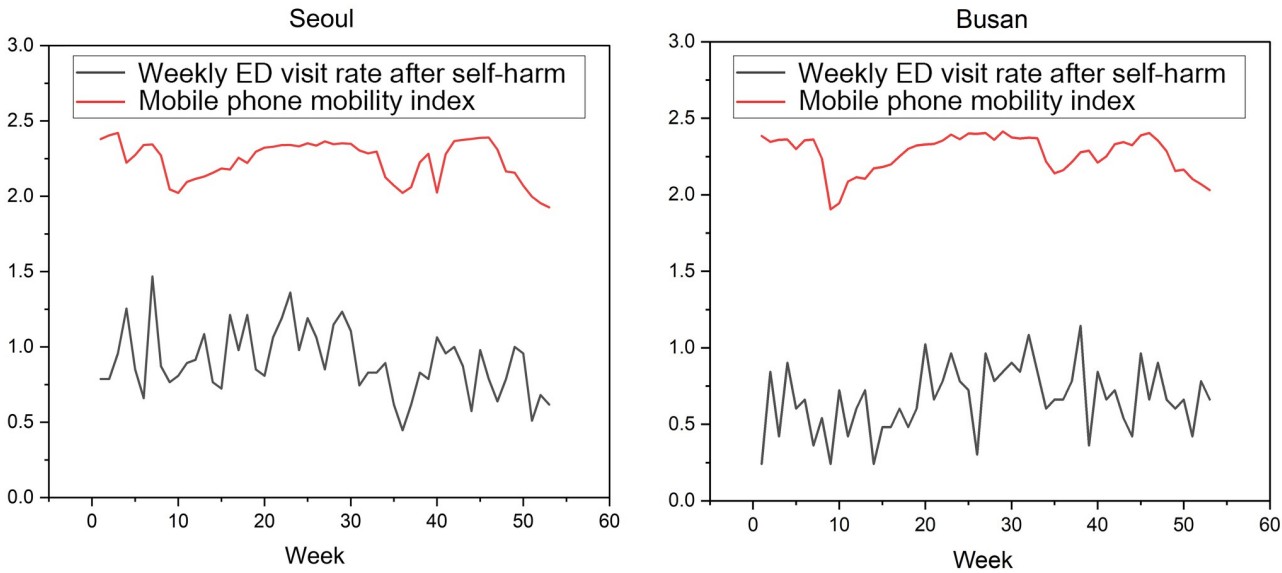

**Fig 2. In Seoul and Busan (urban regions), the correlation coefficients between the mobile phone mobility index and the weekly emergency department visit rate after self-harm were 0.659 and 0.693, respectively, with lag times of 0 weeks and 1 weeks.**

which is slightly shorter than 0.500 (IQR: 0.000–1.250) weeks seen in urban regions, but not statistically significant (p = 0.245).

## Discussion

Proportions of women and young people in self-harm patients increased during the early COVID-19 pandemic compared with five years prior to the pandemic, as well as the

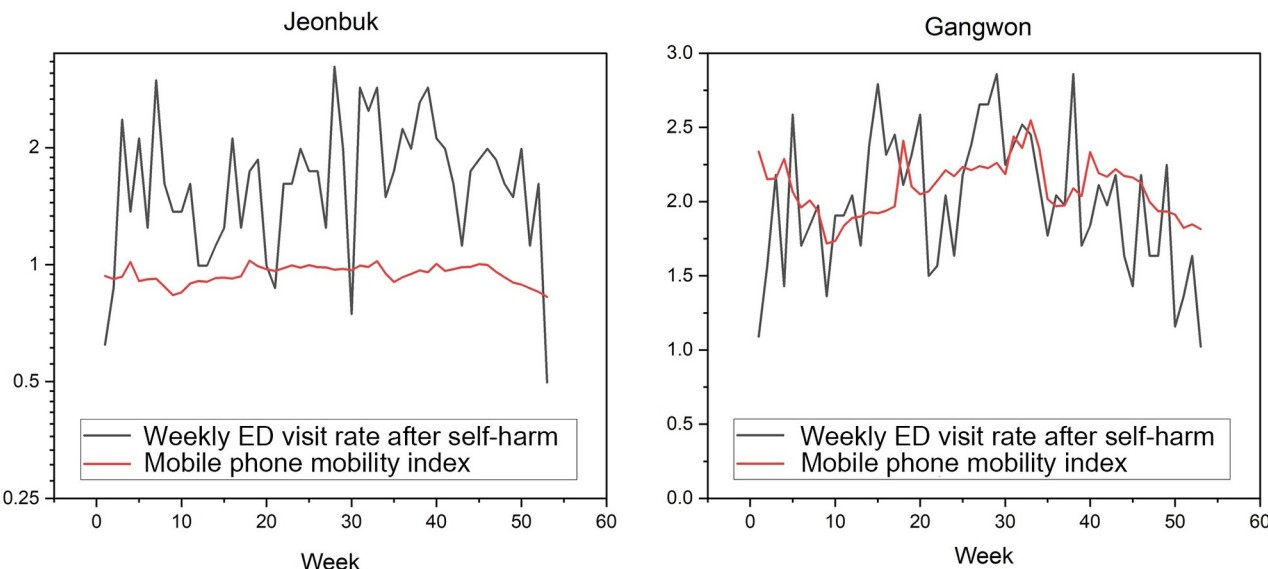

**Fig 3. In Jeonbuk and Gangwon (rural regions), the correlation coefficient between the mobile phone mobility index and the weekly emergency department visit rate after self-harm were 0.653 and 0.682, respectively, with lag times of 0 week and 3 weeks.**

**Table 4. Results of t-tests comparing mean correlation coefficient and lag time between two regions (urban and rural).**

| Variable | Population targeted | Urban [median (IQR)] | Rural [median (IQR)] | Shapiro-Wilk W (p-value) | P-value* |
|---|---|---|---|---|---|
| Correlation coefficient | Total population | 0.601 (0.539–0.619) | 0.531 (0.454–0.595) | 0.851 (0.099) | 0.092 |
| | Male population | 0.504 (0.411–0.615) | 0.425 (0.357–0.519) | 0.937 (0.320) | 0.156 |
| | Female population | 0.451 (0.397–0.629) | 0.433 (0.279–0.504) | 0.933 (0.280) | 0.495 |
| Lag time (weeks) | Total population | 0.500 (0.000–1.250) | 0.000 (0.000–0.250) | 0.591 (<0.001) | 0.245 |
| | Male population | 0.000 (0.000–1.000) | 0.000 (0.000–1.750) | 0.690 (<0.001) | 0.146 |
| | Female population | 0.500 (0.000–2.000) | 0.500 (0.000–1.000) | 0.570 (<0.001) | 0.819 |

IQR; inter-quartile range,

* A Shapiro-Wilk test was performed to test for normality of the data and a student t-test was performed if the normality test passed (p > 0.050), otherwise, a Mann-Whitney U test was conducted.

proportion of self-harm patients in urban regions increased. According to self-harm methods, stabbing or lacerations with sharp objects increased, while hanging and fall decreased significantly. The main purpose of this study was to analyze the relationship between MPMI values (an indicator of physical distancing) and ED utilization resulting from self-harm during the early COVID-19 pandemic period. ED visits after self-harm during the early COVID-19 pandemic declined significantly. Implementing physical distancing measures decreased ED VRSH rapidly within a week in both urban and rural regions.

Several adverse mental health effects might be associated with communicable disease pandemics, including 1) concerns about the risk of infection; 2) loss of social contacts; 3) emotional distress such as worry and anxiety for close acquaintances affected by the disease, and 4) barriers to mental health management caused by lockdown [21]. A study conducted in Chinese mainland located in East Asia like South Korea has reported that the number of nonsuicidal self-injuries was increased significantly (adjusted odds ratio = 1.35 [95% CI: 1.17, 1.55]) in a student cohort during the COVID-19 lockdown [22]. However, another study has reported that a stay-at-home order is associated with a decrease in suicide ideation and suicide attempts [23]. According to a Canadian study, the rate of self-harm or overdose was significantly lower during the pandemic than that before the pandemic [24]. Researchers have speculated that banding together during a pandemic might have contributed to the decline in self-harm and suicide attempts during the COVID-19 pandemic [23, 24]. However, most previous studies have relied on survey data before and after the COVID-19 pandemic or before and after stay-at-home orders without analyzing quantitative indices.

After the implementation of physical distancing measures in accordance with the pandemic, ED attendance due to self-harm dropped by 5.9% compared to that in the previous year in this study. Studies published in several countries reported similar results. In European studies, ED presentations for self-harm and suicide attempts decreased by 13.5%, 42.6%, and 42.0% following the implementation of lockdown restrictions [25–27]. A survey study conducted in New Zealand also revealed a reduction in suicide attempts by 2.1% (95% CI: 1.5%–2.9%) during the pandemic period [28]. A study from Western Australia reported that ED presentations after suicidal behavior and self-harm decreased by 26% during the COVID-19 pandemic [29]. According to studies conducted in the US, Italy, Ireland, Greece, and Australia, the number of patients seeking treatment for self-harm also decreased [27, 29–32].

In contrast, a study conducted in China indicated that mandatory quarantine status was positively associated with self-harm/suicidal ideation [33]. There was a significant increase in self-harm presentations to level one trauma centers in the United Kingdom (UK) in 2020 compared to the same period in 2019 [34]. A study conducted in the US reported an increase in

suicide attempts, including suicide attempts by jumping [30]. However, it is relevant to note that the sample sizes of these studies were small, and they were limited to the very early stages of the pandemic [28, 35, 36]. Therefore, our study attempted to obtain results by analyzing data from all EDs in the country.

The results of this study showed that the proportion of young people among self-harm patients and VRSH in young people increased significantly compared to the pre-pandemic period. The proportion of young people between the ages of 25 and 34 increased to its highest level ever, consistent with a previous study [29]. A previous review also reported that the pandemic and lockdown measures had a significant impact on mental health in young people, including anxiety, stress, depression, event-specific distress, and decreased psychological well-being [37]. Other studies from Asia and Europe found that young people experienced depression symptoms during the COVID-19 outbreak [38–41]. In the field of education, most young people became depressed as a result of concerns regarding their academic performance and forced scholarship termination [38]. The long-term quarantine due to the COVID-19 pandemic may have exacerbated psychological stress and decreased the learning performance of young students [42]. Hence, in the pandemic era of infectious diseases, it is necessary to focus on activities to promote mental health and reinforce social relationships among young people.

Previous studies conducted during the COVID-19 pandemic period found that women experienced more psychological problems such as stress, depression, and self-harm than men [41, 43]. Asian studies have also found that women are more likely to experience psychological effects of pandemics than men [44–47]. These studies postulated that women's susceptibility to self-harm might be due to fears of infection and morbidity, difficulties in broader social communication, and lack of information [34]. They claimed that it is essential to pay increased attention to women's mental health during the pandemic. The proportion of females among self-harm patients increased significantly compared to previous years in this study. An increase in population standardized VRSH was also observed during the pandemic period compared to previous years, although this increase was not statistically significant. Based on this finding, psychological evaluation and support strategies for women who are vulnerable to social isolation need to be continuously implemented. On the other hand, a study conducted in England during the COVID-19 pandemic reported that the ratio of female-to-male presentation changed from 2.0:1 prior to the lockdown to 1.5:1, indicating a greater reduction in females than in males [25], although the difference was not statistically significant. Further studies are needed to validate this sexual difference in consequences of lockdown.

In self-harm methods, the proportion of patients injured by sharp objects increased, while the number of patients injured by hanging and falling decreased significantly. Similar findings were reported in the UK [48]. Continuous attention is required to prevent injuries using tools readily accessible at home in the event of a lockdown.

MPMI values were used as a dynamic measure of physical distancing in this study. Previous studies suggested that mobile phone mobility was useful for observing the dynamics of physical distancing behaviors related to COVID-19 spread [49–51]. A study conducted in Japan reported that the mobile phone mobility of an infection-risk-exposed group was associated with the number of infections with a lag time of 0–2 days, similar to the current study [52]. They used a cross-correlation function to examine the association between the moving average of the mobility index and the time-lagged daily infected cases. The study only examined mobility data for a location that approximates a recreation center, while this study collected weekly mobility data nationwide.

Previous study have evaluated risk groups and urban/rural disparities and concluded that rural residents are more likely to experience psychological distress due to the COVID-19 pandemic [53]. However, this study found the proportion of self-harm patients in urban regions

increased significantly compared to those in previous years. VRSH reduction after strengthening physical distancing measures occurred rapidly in both urban and rural regions in this study. South Korea commercialized code division multiple access for the first time in the world in 1996, constructed the world's first commercial 5G network, and built a tight information and communication network within a narrow perimeter, enabling users to access high-speed internet systems wherever they are. A previous study in Korea found that the rural-urban division was not as evident in terms of digital access and that the frequency of online engagement or the amount of time spent on the web did not differ between the two regions [54]. Consequently, the quality of information did not differ between urban and rural regions. Psychological effects of physical distancing were approximately the same. Therefore, it appears that information systems should be developed and made available to everyone equally to promote social closeness during a pandemic, even if physical distance is maintained.

This study found that 93.5% of self-harm patients visited the ED directly during the early pandemic period, which was a significant decrease from previous years (99.7–99.8%). In contrast, the proportion of patients referred to the ED from outpatient departments rose from 0.1% to 6.2%. Self-harm patients with mild symptoms may have overestimated the risk of COVID-19 infection by visiting the ED and thus failed to seek emergency care. Previous studies reported similar findings that ED utilization decreased significantly following mild trauma or injury [55–57]. However, it should be noted that the proportion of patients with alert mental status at the time of ED arrival who had mild injuries increased significantly (70.4%, p < 0.001) compared to previous years in this study. This indicates that neglecting to get ED treatment after mild self-harm injury is unlikely to occur. Proportion of unresponsive patients (17.4%) also increased significantly compared with previous years. Patient's mental state was biased toward alert and unresponsive. A study published in Ontario, Canada, which offers universal medical insurance in a similar fashion to Korea, also indicated a similar trend [58]. They explained that the decrease in emergent (triage level 2) and urgent (triage level 3) patients was due to the fear of COVID-19 infection in emergency rooms. It postulated that lower-acuity patients do not seek timely care, so they switched to high-acuity patients when they arrived at the ED. Consequently, we also hypothesized that the acuity of patients in this study was polarized in the same way. However, a study conducted in Italy during the early pandemic period found only a decrease in lower-acuity patients [59]. Accordingly, the proportion of patients with lower- or high-acuity patients may vary based on the duration of the pandemic, the number of confirmed cases, and the reporting attitude of the media. Therefore, the implementation of measures that could monitor these changes in real time and countermeasures is necessary.

In this study, large-scale nationwide data on ED surveillance were analyzed and compared to previous years' data. However, there were some limitations to this study. These limitations included the fact that it was conducted in South Korea with a COVID-19 incidence rate that remained low throughout the study period, unlike in Western countries. And bias may be exacerbated by South Korea's highest suicide rate in the world. In this study, MPMI was used as an independent variable to measure social distance changes, not actual numbers of confirmed cases, since the fear of new transmissible diseases and the stress caused by disruptions in social relationships will have a greater psychological impact on the public than the number of confirmed cases themselves. More research is needed, especially those that combine multiple variables, such as confirmed cases and a quantitative measure of physical distancing, to clarify the generalizability of the results. Finally, the study period was only one year, whereas that of the control group was five years, which may have resulted in unbalanced results.

The physical distancing measures adopted to prevent transmittable diseases spread following a pandemic had the effect of decreasing EMSs utilization due to self-harm. The proportion of young self-harm patients increased, but the proportion of older self-harm patients

decreased, and the population standardized VRSH was also found to increase in younger aged population. Women were more likely to self-harm during the pandemic than men. Within one week, the ED VRSH decreased as the level of physical distancing increased. There was no difference in this trend between urban and rural regions. Upon recovery from the pandemic, when daily life is restored to previous levels, it is especially imperative to remain in contact with those at risk of self-harm and maintain social connections.

## Supporting information

**S1 Fig. Graph showing the joinpoint regression model for annual visit rate after self-harm (VRSH) from 2015–2020 with joinpoint at the end of 2019.** The weighted average of slope coefficients measured as average annual percent change (AAPC) was -1.77 between 2015 and 2012 (blue line). And annual percent change (APC) was -11.98 between 2019 and 2020 (green line). T-statistics were calculated to verify APC and AAPC slopes in a straight linear relationship (null hypothesis). We verified that the difference between two slopes was significantly different from zero at alpha = 0.050 level. The null hypothesis was rejected since p-value was 0.014 in this joinpoint plot. In terms of results, the APC had changed significantly at the end of 2019.
(PDF)

**S2 Fig. Cross-correlation plot between weekly mobile phone mobility index values and weekly emergency department visit rates after self-harm.** Cross-correlation function measures the relationship between a time series and a lag version of another time series. This plot showed a maximum correlation coefficient of 0.612 and a lag time of zero weeks.
(PDF)

## Acknowledgments

We would like to thank Soo Young Kim (Professor, Department of Preventive Medicine, College of Medicine, Eulji University, Daejeon, Republic of Korea) for his help in analyzing joinpoint regression.

## Author Contributions

**Conceptualization:** Ye Ji Lee, Seon Hee Woo, Sungyoup Hong.

**Data curation:** Ye Ji Lee, Sungyoup Hong.

**Formal analysis:** In Soo Kim, Sungyoup Hong.

**Investigation:** Ye Ji Lee, Min A. Yuh, Byul Nym Hee Cho, Seon Hee Woo, Sungyoup Hong.

**Methodology:** Min A. Yuh.

**Resources:** In Soo Kim, Byul Nym Hee Cho.

**Software:** Sungyoup Hong.

**Validation:** Byul Nym Hee Cho, Sungyoup Hong.

**Visualization:** In Soo Kim.

**Writing – original draft:** Ye Ji Lee, Min A. Yuh, Sungyoup Hong.

**Writing – review & editing:** Seon Hee Woo, Sungyoup Hong.

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
