## [Decision Letter · Decision Letter 0]

24 Jan 2023

PONE-D-22-22495Physical distancing and emergency medical services utilization by self-harm patients in Korea during the early COVID-19 pandemic: A nationwide quantitative studyPLOS ONE

Dear Dr. Hong,

Thank you for submitting your manuscript to PLOS ONE. After careful consideration, we feel that it has merit but does not fully meet PLOS ONE’s publication criteria as it currently stands. Therefore, we invite you to submit a revised version of the manuscript that addresses the points raised during the review process.

The manuscript has been evaluated by three reviewers, and their comments are available below.

The reviewers have raised a number of concerns that need attention. They request additional information on methodological aspects of the study, and revisions to the statistical analyses.

Could you please revise the manuscript to carefully address the concerns raised?

We look forward to receiving your revised manuscript.

Kind regards,

Steve Zimmerman, PhD

Associate Editor, PLOS ONE

Journal Requirements:

Reviewers' comments:

Reviewer's Responses to Questions

**Comments to the Author**

1. Is the manuscript technically sound, and do the data support the conclusions?

Reviewer #1: Yes

Reviewer #2: Partly

Reviewer #3: Partly

2. Has the statistical analysis been performed appropriately and rigorously? 

Reviewer #1: Yes

Reviewer #2: No

Reviewer #3: No

3. Have the authors made all data underlying the findings in their manuscript fully available?

Reviewer #1: Yes

Reviewer #2: Yes

Reviewer #3: Yes

4. Is the manuscript presented in an intelligible fashion and written in standard English?

Reviewer #1: No

Reviewer #2: No

Reviewer #3: Yes

5. Review Comments to the Author

Reviewer #1: This is a very important manuscript of the secondary effects of Covid-19 pandemic on peoples' mental health and self harm. I only have a few comments.

You could start the Results section byt one or two general sentence before jumping in to Table 1.

Table 2 Mentality upon arrival to ED: In 2020 the patients seems to be more polarized into two groups alert and unresponsive. Could you discuss why is that.

Discussion: You now start this section telling what was the purpose of the study. This belongs to the end of the introduction and to the Methods. I suggest that you would focus to the main findings of your study in the first paragraph of the discussion.

Reviewer #2: The topic of this article is interesting and it contributes to research on COVID-19 by analysing changes in the utilization of emergency services among people after self-harm injury during COVID-19 pandemic.

There are a some issues with the methods and analysis that need to be clarified. The article requires also some language quality check.

Below are more specific comments on areas that need revision:

Introduction:

1.)The authors reference other studies that have explored emotional distresses as a result of COVID-19, but they do not refer to any studies from Korea. More description of the situation in Korea during pandemic and the lockdown would be helpful. It could help to get inside to how the population in Korean reacted to the pandemic.

2.) Line 2 and 4 at the end of the sentences are missing references

3.) Section 3 after the sentence”…. research has found that peer connections confer a level of protection against suicide attempts in a variety of conditions.” is missing reference

4.) In the aim section the authors did not mention (but mention it in the abstract) that they compare the characteristics of self-harm between urban and rural region in the early pandemic (2020).

Methods

1.) Line 4 The authors use incorrect word "mechanism". The correct word would be method of self-harm, which also is used in all theoretical literature and empirical research.

Measurements

2.) Interpretation of incidence rate (IR) is wrong as the denominator has to represent total time at risk (i. e., total person-time). Person-time takes into account the number of people in the group and their time at risk. By calculating weekly incidence rate, the authors should divide by 52. Otherwise the authors should explain/discuss it.

Statistical analysis

3.) The correct terminology is self-injury method or self-harm method, not injury mechanism. Mechanism refers to something else. Mechanism is why people contemplate or self-harm or what makes people have the this kind of behaviour.

Results

1.) Be consistent in choosing urban/rural area or urban/rural region. What is the proportion of these 2 areas? There is a missing information on percentage of patients in urban and rural areas. This is important information as your study is based on this.

2.) If authors are doing CHI-square analysis, than write Chi values with the P values.

3.) Line 4: "…poisoning was the most prevalent cause". Poisoning is a method, not a cause. Cause has a different meaning. Cause of self-harm can be, for example breakup or depression.

4.) Correct use of word is poisoning not poisonings.The same problem is with other methods.

5.) Line 5 and table 1: In text authors use hanging, in table 1 the authors use choking. Hanging and choking are two different methods. It is either hanging or choking or hanging and choking.

6.) Line 6: "…who visited the ED through outpatient units was high in rural areas". Don’t the authors mean higher in rural than urban areas?

7.) Table 1 : check the numbers and percentages again in Table 1

Struck 822 (4.0) 280 (3.3) 1197 (3.9) 822/22421*100=3.67

here should be 3.7

Chocking 893 (3.7) 304 (3.6) 1102 (3.6) 893/22421*100=3.98

here should be 4.0

Beside 304 is wrong number:

1102-893=209 and not 304

8.) Text under the table 1: what kind of test are the p-values obtained from? Is it Chi-squared test?

9.) Be consistent in using self-harm or self-injury. In the title the authors use self-harm, then the word should be self-harm.

10.) 1 sentence in section under the table: there is something wrong with this sentence "……was higher than in 2019 (+2.35%) than during the same period in 2019".

Changes in the incidence of self-harm during the early stages of the pandemic

1.) Section 2: In this sentence: "Compared to the previous five years, the proportion of self-harm patients aged 15 to 34 increased significantly in 2020 (P< 0.046)…”. - You cannot merge 2 age groups to one age group when you have 2 age groups with different P-values in the Table 2. 15-24 and 25-34 (p<0.001 and 0.046, respectively).

”…. whereas those aged 35 and older decreased significantly (P < 0.050)". Same problem here. P-values are different for each age group. You can write (Table 2).

2.) Section 2 sentence 4: delete the incidence of stubbing injuries; The authors use incorrect wording of incidence of stubbing : They do not calculate incidence of stubbing. The correct word is self-harm by stabbing;

3.)Again methods, not causes

4.)Section 2 sentence 5: you cannot merge all methods with one p-value, when the p-values are different for each method

5.)Section 2 sentence 7: During the pandemic era, 93.5% of patients walked directly to the ER, compared to over 99% in the past (P < 0.001). If you round the number in the past, then you should also round the number during the pandemic. Beside it is almost 100% when rounding off the decimal numbers. You can write (99.7 - 99.8%) instead.

Discussion

The authors should discuss that fear of being infected with corona can also lead to reduction in ED visits.

Reviewer #3: Summary:

This study aimed to compare the incidence, proportion, demographic and other clinical characteristics of ED visits for self-harm between rural and urban regions in South Korea, during 2020 and 2015-2019. The authors found an overall decrease in ED visits for self-harm in 2020 compared to previous years, which was correlated with mobile mobility measures (a measure of social distancing) suggesting social distancing practices led to a decrease in usage of ED services. They also report ED visits for self-harm exhibited higher proportions of females and young adults in 2020 compared to previous years. While the aims of this study are important and worth investigating, significant revisions to the methodological and statistical approaches are required to accurately produce and interpret the findings.

Introduction

● The introduction states “the study intended to verify that the utilization of emergency care after self-harm increased as physical distancing measures were implemented”. Is there more background information to support why the authors made this hypothesis? Most previous literature show decreased use of emergency services due to COVID-19 precautions (including the results of this study!).

● One of the major comparisons is between individuals from rural and urban regions. Could the authors present more background on expected differences between these regions?

Methods

● Measurements: could the authors differentiate between accidental self-harm from intentional self-injury? Intentional-self harm can also occur without suicide intent and this is quite different from accidental self-harm that may not have a significant psychological component. Since one of the goals of this study is to determine how to allocate mental health services to prevent intentional self-harm, it would be better to exclude cases of accidental self-harm from the analyses (or compare the results with and without excluding them).

○ Mobile mobility index - What distance away from the home site would count as one mobile mobility point?

● When comparing urban vs. rural areas, what time period was selected as “early pandemic”? Did this period correspond to significant enforcement of social distancing measures?

● Statistical analyses:

○ Two-sample t-tests (comparing rural and urban regions): p-values will need to be adjusted for multiple testing.

○ Chi-square test: omnibus chi-square analyses were done for measurements like injury mechanism and indicate overall significant differences in proportions between regions. However, post-hoc chi-square tests may be required to identify individual levels that significantly differ (for example - is choking really significantly less frequent in rural compared to urban areas, or is the difference in poisoning and stabbing driving the omnibus significance?) (1). Of course, adjustments for p-values must also be done to correct for multiple comparisons.

○ One-sample t-tests: A one-sample t-test was used to compare measurements from the early pandemic and the previous 5 years. Were there 5 separate t-tests for each measurement or was there one t-test comparing the value in 2020 vs the average of the past 5 years?

○ Correlation-shift-function: could the authors provide more details about the parameters selected using this tool? For example, did the authors perform linear or circular correlations? As well, could the authors provide citations for other papers that have used this software for similar analyses?

■ What were the effects of selecting other lag-times? How significantly different were the correlations and do these impact confidence in the interpretations?

● How many urban and rural areas were there?

Results

● Table 1 - general trends in proportions are described but post-hoc statistical tests need to be performed to conclude specific differences such as: In urban areas, the proportion of patients appearing at EDs in an alert mental state was higher than in rural areas (P = 0.046).

○ Here, the p-value is the omnibus chi-square test, not the significance of specific comparisons for “alert mental state” between rural vs urban areas.

● SF1 - general trends were described for this figure. However, real statistical tests are required to make conclusions about differences in mobile mobility each week.

○ This data can actually be used to define specific time-periods of low mobile mobility. It would be interesting to compare intentional self-injury between these time-windows.

○ What does the line represent?

○ What is the mobile mobility ratio? What two values are used to make the ratio?

● Table 2 - unclear what tests the p-values for each row represent. If these are for t-tests, which years were specifically compared? The results section discusses differences in proportions for some of these results, suggesting chi-square tests were done. If so, why is there a p-value for each row?

○ For falling and machine related injuries, were these accidents or purposeful self-harm? If these were accidental, it would be better to remove them from the analyses…

● Table 3 - again, was data from 2020 compared to each of the previous years or an average from the previous years? What was the standard deviation for the 2020 data? Were the units weekly incidence of self harm per 100 k people?

● T-tests need to be corrected for multiple comparisons for both table 1 and 3.

● Fig 1, Fig 2, Table 4:

○ The correlation seems extremely high compared to other studies (2). Why is it so high?

○ How do other values of time-lags affect the correlations?

○ How can the average correlations and time-lags be lower in males and females compared to the overall population? Are they statistically lower (you could use a one-sample t-test here).

Discussion

● The results suggest an increasing trend in self-injury in young people 15-25 over in 2020 compared to previous years. However, there was also a large increase from 2017-2018 in this age group. Is it possible that other factors are contributing to this increase besides the COVID-19 restrictions?

● The results also suggest an increase in incidence and proportion of self-injury in females in 2020 possibly due to COVID-19 restrictions. However, an even larger increase seemed to have occurred between 2017-2018. Since it is unclear if the p-value corresponds to an omnibus test or compared to the average of all years prior to 2020, I’m not sure the authors can make conclusions about incidence.

○ While the proportion of females seemed to have increased in 2020, the interpretation may be very different depending on if the incidence is different or not.

● The authors note that neglecting to get ED treatment after mild self-harm injury is unlikely to occur based on the proportion of alert mental status patients. Can the authors explain this interpretation more? Given the proportion of self-harm ED was lower for older-adults, it's possible that older patients who exhibited intentional self-harm may have perceived the risk of acquiring COVID-19 to be greater than the possible consequences of mild intentional self-harm and chose not to go to the ED. If this was the case, then mental health services should be allocated to older adults as well.

1. Sharpe D. Chi-Square Test is Statistically Significant: Now What? Pract Assess Res Eval [Internet]. 2019 Nov 25;20(1). Available from: https://scholarworks.umass.edu/pare/vol20/iss1/8

2. Kato H, Takizawa A. Time series cross-correlation between home range and number of infected people during the COVID-19 pandemic in a suburban city. PLOS ONE. 2022 Sep 1;17(9):e0267335.

6. PLOS authors have the option to publish the peer review history of their article (what does this mean?). If published, this will include your full peer review and any attached files.

Reviewer #1: No

Reviewer #2: No

Reviewer #3: No

---

## [Author Response · Author response to Decision Letter 0]

22 Feb 2023

Dear Dr Zimmerman,

Thank you for giving me the opportunity to submit a revised draft of my manuscript titled “Physical distancing and emergency medical services utilization after self-harm in Korea during the early COVID-19 pandemic: A nationwide quantitative study” We appreciate the time and effort that you and the reviewers have dedicated to providing your valuable feedback on my manuscript. We are grateful to the reviewers for their insightful comments on our paper. We have been able to incorporate changes to reflect most of the suggestions provided by the reviewers. We are sending the ‘manuscript with track changes’ file along with the ‘flattened manuscript’.

Here is a point-by-point response to the reviewers’ comments and concerns.

Comments from Reviewer 1

General impression: This is a very important manuscript of the secondary effects of Covid-19 pandemic on peoples' mental health and self-harm. I only have a few comments.

Please accept my sincere thanks for reviewing our paper. I have responded to each of the comments below, so please read them and review the manuscript.

▶Comment 1: You could start the Results section byt one or two general sentence before jumping in to Table 1.

▶Response: Thank you for your comments and head of the results section has been revised to include a summary of the study findings. 

▶Comment 2: Table 2 Mentality upon arrival to ED: In 2020 the patients seems to be more polarized into two groups alert and unresponsive. Could you discuss why is that.

▶Response 2: The contents associated with the finding have been added to the upper paragraph of page 16. It would be appreciated if you could review and provide comments.

▶Comment 3: Discussion: You now start this section telling what the purpose of the study was. This belongs to the end of the introduction and to the Methods. I suggest that you would focus to the main findings of your study in the first paragraph of the discussion.

▶Response 3: We changed the head of discussion; I would appreciate you rating it.

Comments from Reviewer 2

Please accept my sincere thanks for reviewing our paper. I have responded to each of the comments below, so please read them and review the manuscript.

Introduction

▶Comment 1: 1) The authors reference other studies that have explored emotional distresses as a result of COVID-19, but they do not refer to any studies from Korea. More description of the situation in Korea during pandemic and the lockdown would be helpful. It could help to get inside to how the population in Korean reacted to the pandemic.

9. Kim HJ, Meeker TJ, Tulloch IK, Mullins J, Park J-H, Bae SH. Pandemic fatigue and anxiety sensitivity as associated factors with posttraumatic stress symptoms among university students in South Korea during the prolonged COVID-19 pandemic. International Journal of Public Health. 2022;67:1604552.

10. Suh BD, Kwon KH. Impacts of the depression among the elderly in the South Korea community in COVID-19 pandemic. Journal of Health Informatics and Statistics. 2021;46(1):54-63.

▶Response: Thank you for your valuable comments. We cited Korean studies that described the effects of pandemic and the lockdown measures for young and elderly adults.

▶Comment 2: Line 2 and 4 at the end of the sentences are missing references

▶Response: Thank you for advice. We added references.

▶Comment 3: Section 3 after the sentence”…. research has found that peer connections confer a level of protection against suicide attempts in a variety of conditions.” is missing reference.

▶Response: Please review the references we cited references emphasize social connectness and friendship network to prevent self-harm and suicide attempts.

▶Comment 4: In the aim section the authors did not mention (but mention it in the abstract) that they compare the characteristics of self-harm between urban and rural region in the early pandemic (2020).

▶Response: Based on your point of view, I have revised the introduction section, please reevaluate it.

Methods

▶Comment 5: Line 4 The authors use incorrect word "mechanism". The correct word would be method of self-harm, which also is used in all theoretical literature and empirical research.

▶Response: As you pointed out, I have revised it accordingly. Thank you. I also modified the same term in table 4.

▶Comment 6: Measurements Interpretation of incidence rate (IR) is wrong as the denominator has to represent total time at risk (i. e., total person-time). Person-time takes into account the number of people in the group and their time at risk. By calculating weekly incidence rate, the authors should divide by 52. Otherwise, the authors should explain/discuss it.

▶Response: Your point about incidence is correct. This term cannot be used in this study without investigating the number of occurrences in a risk group or population group. Therefore, we changed the incidence rate to the “ED visit rate after self-harm”. The word "incidence" has been removed. Please re-evaluate the manuscript.

▶Comment 7: Statistical analysis The correct terminology is self-injury method or self-harm method, not injury mechanism. Mechanism refers to something else. Mechanism is why people contemplate or self-harm or what makes people have the this kind of behavior.

▶Response: We agree with you. All "mechanisms" have been changed to the "self-harm method".

▶Comment 8: Results Be consistent in choosing urban/rural area or urban/rural region. What is the proportion of these 2 areas? There is a missing information on percentage of patients in urban and rural areas. This is important information as your study is based on this.

▶Response: I appreciate you correcting the critical error. Due to the relatively large size of the provinces which we divided, we consolidated them into “region” and revised them.

▶Comment 9: Results If authors are doing CHI-square analysis, than write Chi values with the P values.

▶Response: We added chi-square value along with P-value, additionally post-hoc results. Please review again.

▶Comment 10: Results. Line 4: "…poisoning was the most prevalent cause". Poisoning is a method, not a cause. Cause has a different meaning. Cause of self-harm can be, for example breakup or depression.

▶Response: We are very sorry for the incorrect terminology and grammatical errors. We modified them.

▶Comment 11: Results 4.) Correct use of word is poisoning not poisonings. The same problem is with other methods.

▶Response: We apologize for the grammar error and have corrected them.

▶Comment 12: Results Line 5 and table 1: In text authors use hanging, in table 1 the authors use choking. Hanging and choking are two different methods. It is either hanging or choking or hanging and choking.

▶Response: 'Hanging or chocking' is the correct term. So we abbreviated to 'Hanging/chocking'.

▶Comment 13: Results Line 6: "…who visited the ED through outpatient units was high in rural areas". Don’t the authors mean higher in rural than urban areas?

▶Response: Thank you for the detailed review. I modified it.

▶Comment 14: Results Table 1 : check the numbers and percentages again in Table 1

Struck 822 (4.0) 280 (3.3) 1197 (3.9) 822/22421*100=3.67

here should be 3.7

Chocking 893 (3.7) 304 (3.6) 1102 (3.6) 893/22421*100=3.98

here should be 4.0

Beside 304 is wrong number:

1102-893=209 and not 304

▶Response: Struck (4.0%) and chocking (3.7%) proportion values have been switched. Therefore, we corrected.

▶Comment 15: Results Text under the table 1: what kind of test are the p-values obtained from? Is it Chi-squared test?

▶Response: We stated '* Chi-squared test P-values.’ in table 1 footnote. Moreover, following the advice of other reviewers, we conducted a Chi-square post-hoc test and included the results. Please review it again.

▶Comment 16: Results Be consistent in using self-harm or self-injury. In the title the authors use self-harm, then the word should be self-harm.

▶Response: Thank you. We unified the term.

▶Comment 17: Results 1 sentence in section under the table: there is something wrong with this sentence "……was higher than in 2019 (+2.35%) than during the same period in 2019".

Changes in the incidence of self-harm during the early stages of the pandemic

▶Comment 18: Results Section 2: In this sentence: "Compared to the previous five years, the proportion of self-harm patients aged 15 to 34 increased significantly in 2020 (P< 0.046)…”. - You cannot merge 2 age groups to one age group when you have 2 age groups with different P-values in the Table 2. 15-24 and 25-34 (p<0.001 and 0.046, respectively).

▶Response: We have revised. Thank you.

▶Comment 19: Results ”…. whereas those aged 35 and older decreased significantly (P < 0.050)". Same problem here. P-values are different for each age group. You can write (Table 2).

▶Response: We have revised. Please review again.

▶Comment 20: Results Section 2 sentence 4: delete the incidence of stubbing injuries; The authors use incorrect wording of incidence of stubbing : They do not calculate incidence of stubbing. The correct word is self-harm by stabbing;

▶Response: It was corrected as "the proportion of stabbings increased to 29.5%,"

▶Comment 21: Results Again methods, not causes

▶Response: Thank you so much. That was corrected.

▶Comment 22: Results Section 2 sentence 5: you cannot merge all methods with one p-value, when the p-values are different for each method

Response: Again, we presented all p-values independently.

▶Comment 23: Results Section 2 sentence 7: During the pandemic era, 93.5% of patients walked directly to the ER, compared to over 99% in the past (P < 0.001). If you round the number in the past, then you should also round the number during the pandemic. Beside it is almost 100% when rounding off the decimal numbers. You can write (99.7 - 99.8%) instead.

▶Response: My sincere thanks go out to you for your comments. Updates have been made.

▶Comment 24: Discussion The authors should discuss that fear of being infected with corona can also lead to reduction in ED visits.

▶Response: We have discussed on page 16, section “The results of this study found that ~ And the implementation of measures that can monitor these changes in real time and take countermeasures is therefore necessary.”

Comments from Reviewer 3

Summary: This study aimed to compare the incidence, proportion, demographic and other clinical characteristics of ED visits for self-harm between rural and urban regions in South Korea, during 2020 and 2015-2019. The authors found an overall decrease in ED visits for self-harm in 2020 compared to previous years, which was correlated with mobile mobility measures (a measure of social distancing) suggesting social distancing practices led to a decrease in usage of ED services. They also report ED visits for self-harm exhibited higher proportions of females and young adults in 2020 compared to previous years. While the aims of this study are important and worth investigating, significant revisions to the methodological and statistical approaches are required to accurately produce and interpret the findings.

Please accept my sincere thanks for reviewing our paper. I have responded to each of the comments below, so please read them and review the manuscript.

▶Comment 1: Introduction: The introduction states “the study intended to verify that the utilization of emergency care after self-harm increased as physical distancing measures were implemented”. Is there more background information to support why the authors made this hypothesis? Most previous literature show decreased use of emergency services due to COVID-19 precautions (including the results of this study!).

▶Response: One of the major comparisons is between individuals from rural and urban regions. Could the authors present more background on expected differences between these regions?.

▶Comment 2: Methods ● Measurements: could the authors differentiate between accidental self-harm from intentional self-injury? Intentional-self harm can also occur without suicide intent and this is quite different from accidental self-harm that may not have a significant psychological component. Since one of the goals of this study is to determine how to allocate mental health services to prevent intentional self-harm, it would be better to exclude cases of accidental self-harm from the analyses (or compare the results with and without excluding them).

▶Response: We defined the self-harm as “The term "self-harm" refers to nonfatal intentional self-injury or self-poisoning, regardless of the apparent motivation or suicidal intent” in Introduction section. 

▶Comment 3: Methods Mobile mobility index - What distance away from the home site would count as one mobile mobility point?

▶Response: One-mobile phone mobility is recorded when an individual moves from one village (residence site: staying from midnight to 6 a.m.) to another village (the smallest administrative unit in Korea). The MPMI was calculated by aggregating the total mobile phone mobility and dividing it by the mid-population of each provision. MPMI includes movement statistics for all SK Telecom subscribers aged 15 or older on a weekly basis. This is the data that the National Statistical Office of the Republic of Korea collects and discloses to the public.

▶Comment 4: Methods ● When comparing urban vs. rural areas, what time period was selected as “early pandemic”? Did this period correspond to significant enforcement of social distancing measures?

▶Response: Yes, it was. We included the early pandemic period of 2020 in our analysis, and strong distancing regulations were implemented in South Korea during this time period. Measures including a ban on school attendance and a ban on eating in restaurants and cafes were implemented. And the intensity of bans was adjusted to confirmed cases in each province. Therefore, we collected and analyzed the mobile phone mobility index for each province every week.

▶Comment 5: Methods Two-sample t-tests (comparing rural and urban regions): p-values will need to be adjusted for multiple testing.

▶Response: We conducted a t-test with all independent variables in the syntax and analyzed results were corrected using the Bonferroni method. We added the related contents to the methods section.

▶Comment 6: Methods Chi-square test: omnibus chi-square analyses were done for measurements like injury mechanism and indicate overall significant differences in proportions between regions. However, post-hoc chi-square tests may be required to identify individual levels that significantly differ (for example - is choking really significantly less frequent in rural compared to urban areas, or is the difference in poisoning and stabbing driving the omnibus significance?) (1). Of course, adjustments for p-values must also be done to correct for multiple comparisons.

▶Response: Based on your recommendation, we conducted a chisq post-hoc test for each subgroup of Table 1 after conducting the chisq test on the entire table. It would be greatly appreciated if you could review it.

▶Comment 7: Methods Correlation-shift-function: could the authors provide more details about the parameters selected using this tool? For example, did the authors perform linear or circular correlations? As well, could the authors provide citations for other papers that have used this software for similar analyses?

▶Response: We ran the correlation shift app (https://www.originlab.com/FileExchange/details.aspx?fid=466) using Origin Pro (Origin Lab, Northampton, MA), but we suspect the correlation coefficient values were excessively measured. Therefore, we sent queries to OriginLab (You may refer to the bottom of this page for more information about the query: https://www.originlab.com/FileExchange/details.aspx?fid=466), but were not able to receive an answer. Consequently, the authors used R to test the similarity of two curve (MPMI and VRSH) and lag times between the two groups by performing cross-correlation functions in R. Finally, we re-performed statistical analysis R program and have updated all statistical values in the results section. Please review again

▶Comment 8: Methods What were the effects of selecting other lag-times? How significantly different were the correlations and do these impact confidence in the interpretations?

▶Response: I have attached a cross correlation (CC) graph for your reference. While the correlation coefficients of VRSH and MPMI will differ from region to region, the CC analysis will generally produce similar graphs. Besides, the lag time is not selected by the authors. Instead, the R program automatically selects the point where maximum height CC appears while moving VRSH and MPMI left and right. The highest CC may be low and the 95% confidence interval may be wide if you choose a different lag time.

▶Comment 9: Methods How many urban and rural areas were there? 

▶Response: There are six metropolitan cities classified to urban areas. Rural areas are consisted with 10 provinces. At the beginning of the results, relevant content is described.

▶Comment 10: Methods Table 1 - general trends in proportions are described but post-hoc statistical tests need to be performed to conclude specific differences such as: In urban areas, the proportion of patients appearing at EDs in an alert mental state was higher than in rural areas (P = 0.046).

▶Response: Thank you very much for your advice. A chi-square test and a post-hoc test were performed and the results were added to the results section.

▶Comment 11: Results Table 1 - general trends in proportions are described but post-hoc statistical tests need to be performed to conclude specific differences such as: In urban areas, the proportion of patients appearing at EDs in an alert mental state was higher than in rural areas (P = 0.046).

▶Response: I completely agree with you. The chi-square coefficient depends on the strength of the relationship and sample size. It had occurred to me that it would be beneficial to add an analysis to measure this trend from the beginning. We analyzed table 1 again and calculated the contingency coefficient (CC), the most employed method to measure the association of x-y in a table. These are also described in the Results section.

▶Comment 12: Results Here, the p-value is the omnibus chi-square test, not the significance of specific comparisons for “alert mental state” between rural vs urban areas.

▶Response: I apologize for the error. I conducted a Chisq Post-hoc test and attached the results. As you stated, there is no difference in the level of consciousness between the two regions (P>0.243).

▶Comment 13: Results SF1 - general trends were described for this figure. However, real statistical tests are required to make conclusions about differences in mobile mobility each week.

▶Response: Repeated measured ANOVA was performed and the results were added to the results section. S1 Fig. has been changed to Figure 1 and moved into the text. Thank you.

▶Comment 14: Results This data can actually be used to define specific time-periods of low mobile mobility. It would be interesting to compare intentional self-injury between these time-windows.

▶Response: There are 16 provinces in Korea. Local governments adjusted the strength of the social distancing measure based on COVID-19 epidemic status. There may were a change in MPMI as a result.

If you want to analyze only the MPMI reduction time-windows separately, it should be selected separately for each province. It is also critical to define the reduction criteria. In my opinion, there are too many variables to consider.

The authors tried to find relationship between MPMI as a distancing parameter and VRSH using cross-correlation. Because it was most effective way to find CC and lag time mathematically is by moving the curves of MPMI and VRSH for each local region from one side to the other.

I'd appreciate it if you could take this into consideration.

▶Comment 15: Results What does the line represent?

▶Response: It appears that Figure S1 contains a large amount of meaningful data, so I have changed Figure 1. Additionally, the graph lines show the means and error bars present 95% CI for VRSH based (we changed error bars from SEs to 95% confidence intervals). In my opinion, we can realize a clearer picture of the changes because of the lockdown and time period. 

▶Comment 16: Results What is the mobile mobility ratio? What two values are used to make the ratio?

In SF1, the 2020/2019 ratio represents the change in MPMI in 2020 as compared to MPMI in 2019. This graph line shows how MPMI has decreased for each week in 2020 as compared to 2019.

▶Response: Even in normal years, MPMI varies according to the four seasons and the vacation season. When we first tried the analysis, we tried to show the ratio by dividing MPMI in 2020 by MPMI in 2019. As a result, the ratio has been removed, as the values from the two regions have been presented in graphs for each of 2019 and 2020 (Fig. 1). I would appreciate it if you could review this again.

▶Comment 17: Results Table 2 - unclear what tests the p-values for each row represent. If these are for t-tests, which years were specifically compared? The results section discusses differences in proportions for some of these results, suggesting chi-square tests were done. If so, why is there a p-value for each row?

▶Response: It makes sense to use the Chisq test to determine whether the overall distribution varies significantly by group from 2014 to 2020. However, the authors are interested in determining whether there was a significant change in 2020 relative to the previous five years.

Therefore, we compared the proportion (%) in 2020 to the average value of the previous year using a one-sample t-test. Therefore, it is correct that each row has a p-value. And

▶Comment 18: Results For falling and machine related injuries, were these accidents or purposeful self-harm? If these were accidental, it would be better to remove them from the analyses…

The study does not include unintentional injury cases, as "machine-related" and "falling" self-harm are only included if they are confirmed by triage physicians or nurses as self-harm.

The research method indicates that all subjects included in this study are "intentionally self-injured patients". When collecting NEDIS data, all unintentional injuries or traumas are sent as "unintentional" to the server. Consequently, only "purposeful self-harm" was entered into the research target.

▶Response: Absolutely correct. Unintentional accidents are coded with a different value. So only self-harm cases are classified as “intentional self-harm” at ED arrival.

▶Comment 19: Results Table 3 - again, was data from 2020 compared to each of the previous years or an average from the previous years? What was the standard deviation for the 2020 data? Were the units weekly incidence of self-harm per 100 k people?

▶Response: Incidence is wrong term. Therefore, we changed to correct term “weekly ED visit rate after self-harm per 100,000.” in entire manuscript. The SD value could not be obtained for 2020 data because of a single data set. The 2020 values were compared with data from the past five years using a student t-test or a Mann-Whitney U test.

▶Comment 20: Results T-tests need to be corrected for multiple comparisons for both table 1 and 3.

P-value in table 1 is from chisq test. And we did chisq post-hoc test according to your advice and added the results.

▶Response: We used the one sample test comparing the 2020 value with the previous year's average in table 2 and3. A multiple comparison was not conducted. I would appreciate it if you could take this into consideration and again provide me with your advice.

▶Comment 21: Results Fig 1, Fig 2, Table 4: The correlation seems extremely high compared to other studies (2). Why is it so high?

▶Response: We ran the correlation shift app (https://www.originlab.com/fileExchange/ details.aspx?fid=466) using Origin Pro (Origin Lab, Northampton, MA), but we suspect the correlation coefficient values were excessively measured. Therefore, we sent queries to OriginLab, but were not able to receive an answer. Consequently, the authors used R to test the similarity of two curve (MPMI and VRSH) and lag times between the two groups by performing cross-correlation functions. Finally, we have updated all statistical values in the results section. Please review again

▶Comment 22: Results How do other values of time-lags affect the correlations?

▶Response: When we performed cross correlation with R.

Sample data

VRSH <- c(2.09 ,2.03 ,2.04 ,2.25 ,1.99 ,2.02 ,2.04 ,1.85 ,1.64 ,1.68 ,1.78 ,1.84 ,1.84 ,1.93 ,1.91 ,1.92 ,1.97 ,2.13 ,2.07 ,2.05 ,2.07 ,2.07 ,2.10 ,2.05 ,2.09 ,2.09 ,2.09 ,2.06 ,2.09 ,2.05 ,2.16 ,2.17 ,2.21 ,2.10 ,2.00 ,1.98 ,2.03 ,2.09 ,2.07 ,2.22 ,2.09 ,2.11 ,2.12 ,2.12 ,2.13 ,2.13 ,2.05 ,2.01 ,1.98 ,1.95 ,1.87 ,1.85 ,1.80 )

MPMI <- c(0.97 ,0.94 ,0.95 ,1.05 ,0.93 ,0.94 ,0.95 ,0.86 ,0.76 ,0.78 ,0.83 ,0.86 ,0.86 ,0.90 ,0.89 ,0.90 ,0.92 ,0.99 ,0.97 ,0.95 ,0.96 ,0.97 ,0.98 ,0.95 ,0.98 ,0.97 ,0.97 ,0.96 ,0.97 ,0.96 ,1.01 ,1.01 ,1.03 ,0.98 ,0.93 ,0.92 ,0.95 ,0.98 ,0.96 ,1.04 ,0.97 ,0.98 ,0.99 ,0.99 ,0.99 ,0.99 ,0.96 ,0.94 ,0.92 ,0.91 ,0.87 ,0.86 ,0.84 ) 

print(ccf(province, mpmi))

Moving left and right, you can observe a change in CC values as a result of the change in lag time.

Most of the highest correlation occurred around 0-1 week, and the change in correlation according to lag time can be seen in the output data above or the figure below. I would appreciate it if you could refer to them.

▶Comment 23: Results How can the average correlations and time-lags be lower in males and females compared to the overall population? Are they statistically lower (you could use a one-sample t-test here).

▶Response: Thank you for taking the time to review this in detail. In the process of transferring the Origin program output, the CC values and lag time values of the male and female were misrepresented as 0.000 (default). We have updated the original data. It would be greatly appreciated if you could review Table 4 and reevaluate it.

We compared CC value and lag time for two groups (Urban areas (7 metropolitan cities) and rural areas (9 provinces) in Korea) by student T-test.

▶Comment 24: Discussion The results suggest an increasing trend in self-injury in young people 15-25 over in 2020 compared to previous years. However, there was also a large increase from 2017-2018 in this age group. Is it possible that other factors are contributing to this increase besides the COVID-19 restrictions?

▶Response: That is a valid point, in my opinion. It is possible that factors such as university entrance examination systems and changes in social welfare may affect self-harm among students aged 15 to 25. There was also a political disruption (transfer of power from the right-wing government to the left-wing government) in 2017. However, the main focus of this study is to compare the 2020 early pandemic period with the previous years. If ED visits after self-harm have increased significantly in the previous years (2017 vs 2018), the one-sample T-test result should show no significance comparing 2020 with previous years. Considering these political factors, it is reasonable to explain this increase as COVID-19 as a result of a significant increase in the 15-25 groups in 2020.

▶Comment 25: Discussion The results also suggest an increase in incidence and proportion of self-injury in females in 2020 possibly due to COVID-19 restrictions. However, an even larger increase seemed to have occurred between 2017-2018. Since it is unclear if the p-value corresponds to an omnibus test or compared to the average of all years prior to 2020, I’m not sure the authors can make conclusions about incidence.

▶Response: That is correct, thank you. In 2017 and 2018, the population standardized visit rate for women increased significantly by 19.29% and 13.69% compared to the previous year. There was, however, a reduction in the increase in 2020 to 7.25%. We believe that lockdowns and social distancing will reduce emergency department visit rates after self-harm in 2020, which supports our argument.

I would appreciate it if you could consider this and review it again.

▶Comment 26: Discussion While the proportion of females seemed to have increased in 2020, the interpretation may be very different depending on if the incidence is different or not.

In Table 2, the proposition of female patients increased significantly in 2020. In addition, it can be seen that the population standardized visibility rate in table 3 also decreases in men, but rather increases in women.

I have concluded that the change in the proportion of females can be explained by the unique increase of population standardized visit rate in the female group.

▶Response: There are some points on which I agree with you. As a result, however, if population standardized VRSH is to be directly related to changes in female VRSH in the entire population, we must assume that sex ratios in all provinces are the same. Because we can't confirm it like that, I don't think we can assure the crude female visit ratio only with population standard VRSH.

Therefore, we expressed it on page 15 as follows. "In this study, the proportion of females among self-harm patients and the population standardized VRSH among females increased significantly from previous years."

In my opinion, it would be appropriate to present both findings. It would be appreciated if you could review it again after taking this into consideration.

▶Comment 27: Discussion The authors note that neglecting to get ED treatment after mild self-harm injury is unlikely to occur based on the proportion of alert mental status patients. Can the authors explain this interpretation more? Given the proportion of self-harm ED was lower for older-adults, it's possible that older patients who exhibited intentional self-harm may have perceived the risk of acquiring COVID-19 to be greater than the possible consequences of mild intentional self-harm and chose not to go to the ED. If this was the case, then mental health services should be allocated to older adults as well.

▶Response: Regarding your concerns, thank you very much. On page 16, we describe the related contents in the second sentence. Patients who self-harm mildly may neglect to visit the ED. Despite that, we attempted to demonstrate that the phenomenon was not significant in this study by citing related studies (reference 56-58). We request that you review this again.

 56. Dowell RJ, Ashwood N, Hind J. Musculoskeletal attendances to a minor injury department during a pandemic. Cureus. 2021;13(2). doi: https://doi.org/10.7759/cureus.13143.

57. Pikoulis E, Koliakos N, Papaconstantinou D, Pararas N, Pikoulis A, Fotios-Christos S, et al. The effect of the COVID pandemic lockdown measures on surgical emergencies: experience and lessons learned from a Greek tertiary hospital. World J Emerg Surg. 2021;16(1):1-8. doi: https://doi.org/10.1186/s13017-021-00364-1.

58. Jacob S, Mwagiru D, Thakur I, Moghadam A, Oh T, Hsu J. Impact of societal restrictions and lockdown on trauma admissions during the COVID‐19 pandemic: a single‐centre cross‐sectional observational study. ANZ journal of surgery. 2020;90(11):2227-31. doi: https://doi.org/10.1111/ans.16307.

1. Sharpe D. Chi-Square Test is Statistically Significant: Now What? Pract Assess Res Eval [Internet]. 2019 Nov 25;20(1). Available from: https://scholarworks.umass.edu/pare/vol20/iss1/8

2. Kato H, Takizawa A. Time series cross-correlation between home range and number of infected people during the COVID-19 pandemic in a suburban city. PLOS ONE. 2022 Sep 1;17(9):e0267335.

▶Response: In the discussion section (page 15), the reference was cited, as well as related findings from this study, including those that differed from the consensus.

▶▶Other amendments.

1. We revised the title of manuscript into “Physical distancing and emergency services utilization after self-harm in Korea during the early COVID-19 pandemic.”

2. We Added missing acronyms to Table footnotes.

3. In the study design sub-section (Methods), it is stated that the study is a retrospective before-and-after study.

4. Additionally, the data related to the COVID-19 confirmed cases were from the time we were submitting the manuscript and had a wide gap from the present, so the data is updated as of December 2022. 

5. We decided that since a province refers to large-scale land, it should be referred to as a region instead of an area. The word 'area' has been changed to 'region'.

6. We thought S1 figure showing significant contents, so it was moved into the text as Figure 1. In the figure there is a comment about the meaning of ratio and what the line represents, so I redesigned it with a more colorful and revised picture. Please review again.

7. We calculated the proportional values of all tables once again and presented them in the unified vertical direction.

8. In Tables 3, five-year sample values make it difficult to guarantee normality. Thus, the first step was to conduct the Shapiro-Wilk test to verify normality, followed by one-sample t-tests for normal distributions and Wilcox signed run tests for non-normal distributions. The statistical method and the p-values have also been updated.

9. In table 4, we also chose a parametric (student t-test) or nonparametric test (Mann-Whitney U test), based on the result of the Shapiro-Wilk test, in order to verify the assumption that the sample distribution corresponds to a normal distribution. We updated our statistical methods and results according to this modification. Along with the statistical methods, we added the Shapiro-Wilk W value and updated the p-value.

10. The normality test results have been added to table 3, 4 as well as the type of statistical analysis used to determine the p-value has been updated in the table footnote.

11. Changes in the methods of self-harm are also significant. Therefore, we included a discussion about the changes in self-harm methods that occurred during the pandemic. 

12. Harrisco & Co. Korea provided us with English editing services again for grammatical and linguistic errors.

---

## [Decision Letter · Decision Letter 1]

14 Mar 2023

PONE-D-22-22495R1Physical distancing and emergency medical services utilization after self-harm in Korea during the early COVID-19 pandemic: A nationwide quantitative studyPLOS ONE

Dear Dr. Hong,

Thank you for submitting your manuscript to PLOS ONE. After careful consideration, we feel that it has merit but does not fully meet PLOS ONE’s publication criteria as it currently stands. Therefore, we invite you to submit a revised version of the manuscript that addresses the points raised during the review process.

We look forward to receiving your revised manuscript.

Kind regards,

Pei Boon Ooi, Ph.D.

Academic Editor

PLOS ONE

Journal Requirements:

Additional Editor Comments:

Thank you for the revision. We would like to invite you to consider and amend the manuscript according to the suggestions given by the reviewers. We look forward receiving the revision.

Reviewers' comments:

Reviewer's Responses to Questions

**Comments to the Author**

1. If the authors have adequately addressed your comments raised in a previous round of review and you feel that this manuscript is now acceptable for publication, you may indicate that here to bypass the “Comments to the Author” section, enter your conflict of interest statement in the “Confidential to Editor” section, and submit your "Accept" recommendation.

Reviewer #1: All comments have been addressed

Reviewer #2: All comments have been addressed

Reviewer #3: (No Response)

2. Is the manuscript technically sound, and do the data support the conclusions?

Reviewer #1: Yes

Reviewer #2: Yes

Reviewer #3: Partly

3. Has the statistical analysis been performed appropriately and rigorously? 

Reviewer #1: Yes

Reviewer #2: Yes

Reviewer #3: No

4. Have the authors made all data underlying the findings in their manuscript fully available?

Reviewer #1: Yes

Reviewer #2: Yes

Reviewer #3: Yes

5. Is the manuscript presented in an intelligible fashion and written in standard English?

Reviewer #1: Yes

Reviewer #2: No

Reviewer #3: Yes

6. Review Comments to the Author

Reviewer #1: The authors have done a good revision and I have no further comments. k

Reviewer #2: Thank you for the revised article. When reading it again, I feel that the article needs major language quality control. There are missing words in the text and the authors mix up past tense with present tense. Please find some (minor) comments below refering to this.

-The authors still use ”area” and ”region” and "province" in the article. Be consistent in using one of these words.

RESULTS

-the first 6 lines in RESULTS belongs to the Methods

-line 6: What time period /year are you referring to in this sentence? "A total of 32,778,938 people lived in the urban regions, and 14,736,844 people lived in the eight rural provinces".

-You wrote "eight rural provinces" but you didn't provide information on how many urban regions there were. You can either delete the 8 or write also number of urban areas.

-line 9: Please rephrase this sentence. The wording is imprecise: "In the chi-square post-hoc test for age-specific

distribution, the proportion in the 15 – 24 and 25 – 34–year–old groups in the rural area, and the proportion in the over–45–year–old group in the urban area increased relative to each other (p < 0.001)."

Furthermore the young (15-24 and 25-34) are overrepresented in urban area according to the table 1, and not rural area, while those aged 45 and over are overrepresented in rural area, and not urban area.

-Also you do not have age group 45+ according to the table 1. You can write instead: those aged 45 and over or patients aged 45 and over.

-line 11: The authors used the past tense and then the present tense. Use past tense in relation to previous text: "Across both study regions, poisoning was the most common method of self-harm, while traffic accidents are..." (WERE)....

-line 11: You are not comparing traffic accidents with laceration/stabbing. Do you mean AND instead of THAN in the sentence?: "Across both study regions, poisoning was the most common method of self-harm, while traffic accidents are more common in rural regions THAN laceration/stabbing..."

-Also I suggest that you place p-values inside the sentence:

…poisoning was the most common method of self-harm (p < 0.001), while traffic accidents are more common in rural regions (p < 0.001), and laceration/stabbing in urban areas (p = 0.006).

-line 13: Accoriding to the table 1,the proportion of patients in outpatient unit was higher in rural than urban regions (41 vs 37): "The proportion of patients who visited EDs through outpatient units was high in urban regions (p < 0.001)".

-line 14: The word WAS is missing in the sentence. Please rephrase this sentence: A higher proportion of patients presented to EDs in an alert mental state WAS in urban regions than in rural regions from chi-square post-hoc test (p<0.001).

-site 8, line 2: There are missing some words in this sentence: "….decreased sharply beginning the first week of February…"

site 9, line 8: Change the word order in this sentence: "The proportion of self-harm patients in two age groups (15 – 24 and 25 – 34) increased significantly in 2020 (p < 0.001 and p = 0.046, respectively), whereas decreased significantly in the 35 – 44, 45 – 54, 55 – 64, 65 – 74, 75 – 84, 85 – 94 and over-95 age groups (p <0.001, p = 0.003, p < 0.001, p = 0.004, p = 0.045, p <0.001 and p = 0.010, respectively)."

-site 9 line 11: Change the word ”and” to "while"in this sentence:

According to the self-harm method only the proportion of laceration/stabbing injuries increased to 29.5%,% (p = 0.006),AND methods such as being struck by an object, hanging/choking, falling, burningfall, submersion, burn, and traffic accidents decreased significantly (p = 0.015, p < 0.001, p = 0.004, p = 0.016, p = 0.004 and p = 0.041 respectively).

-site 11 Table 3: In table 3 the authors calculate annual visit rate after self-harm (VRSH) by age and sex. In the method section, you only describe the weekly VRSH, but do not mention anything about the annual VRSH by age and sex.

-site 11 line 1-2: The authors wrote p=0.35. Do you mean p=0.356?" Female self-harm visits per 100,000 people increased to 78.4 in 2020, showing a sustained increase over the previous five years (p = 0.011), whereas male self-harm visit rates did not significantly increase over time (p = 0.35) (table 3)."

-site 11 line 2: The p-value for group aged 25-34 is <0,001 according to table 3, and not p=0.008. "There was a significant increase in standardized self-harm visit rates for groups aged 15 –24 (p = 0.003) and 25 – 34 (p = 0.008) compared to the previous five years, but a decrease for those 75 – 84 (p = 0.049), 85 – 94 (p = 0.010), 95 years and older (p = 0.011)"

-Also at the end of the sentence “and” is missing before “95 years and older (p = 0.011)”.

-site 12: The first 3 lines belongs to methods.

-site 12 line 9: The word WHICH is missing in this sentence: "In rural regions, the lag time between decreases in the MPMI and VRSH were 0.000 (IQR: 0.000 – 0.250) weeks, WHICH is slightly shorter than 0.500 (IQR: 0.000 -1.250) weeks seen in urban regions, but not statistically significant (p = 0.245)."

Reviewer #3: Review Notes

Comment 1 - the authors did not respond to this comment. The response is simply a copy of one of my other original comments for the introduction. Again, please include what is the motivation for comparing rural vs urban self-harm ED usage in the introduction.

Comment 5 - authors report results were corrected for multiple comparisons using the Bonferroni method and this was included in the methods. Upon review, no mention of Bonferroni correction was reported in the methods section. Please also indicate whether p-values in all the tables are corrected for multiple comparisons in the table legends.

Comment 6 - In the methods, the authors write: “The Student t-test or Mann-Whitney U test was used to compare the characteristics of the two study regions (Table 1)”. Which tests were t-tests/u-tests in Table. 1? Aren’t the tests in Table. 1 all Chi-square tests of independence?

Post-hoc Chi-square tests are only done when contingency tables exceed a 2x2 design. In table 1, they do not need to be done for sex and urban/rural area. Post-hoc Chi-square tests may be done for factors with more than 2-levels, such as the age factor that has 9 levels. Post-hoc tests are typically performed using individual pairs of those levels; ie 9C2 = 36 pairs to test (ex. Ages 15-24 and 25-34 vs. sex, then ages 15-24 and 35-44 vs. sex….etc). Please clarify in the methods and results how the post-hoc Chi-square tests were done. Additionally, corrections for multiple testing will need to be done for the 36 post-hoc chi-square tests for age. Due to the high number of tests, less stringent correction methods such as Bonferroni-Holm can be considered. Here is a resource that provides some guidance on post-hoc Chi-square tests. https://alanarnholt.github.io/PDS-Bookdown2/post-hoc-tests-1.html.

Additionally, if more than 20% of the cells have < 5 counts for each Chi^2 test, Fisher’s exact test should be used instead of Chi-square test of independence.

Comment 10 - see comment 6.

Comment 11 - where are the contingency coefficients (that the authors report they calculated) reported in table 1? They are also not mentioned in the results.

Comment 17 - one-sample t-tests require a sample of observations, and then comparing the data to the expected mean. For the results in Table. 2, the authors state they are comparing the proportion (for example, of females) in 2020 to the average of the previous 5 years. The proportion of females in 2020 is a single observation, not a sample of observations, so it is not possible to do a t-test. How did the authors do t-tests for this data?

When comparing proportions, it is more appropriate to use Chi-square tests of independence. If you have an expected proportion (for example, average proportion of females from 2015 - 2019), you can perform a one-sample Chi-square test by comparing your observed proportion (ex. Proportion of females in 2020) to the expected proportion.

Comment 19 - Table 3 legend indicates “A Shapiro-Wilk test was conducted to test for normality of the variables (2014-2018), and a one-sample t-test was performed if the normality test passed (p > 0.050).” First, the data ranges from 2015-2020, so shouldn’t the authors have conducted Shapiro-Wilk tests on 2015-2019 values? Second, for one-sample t-tests, it seems the authors may have used emergency department rates from 2015-2019 as the “sample” and the emergency department rate in 2020 as the “expected mean” to compare the sample with. If this is the case, then the one-sample t-test is NOT an appropriate statistical test for this analysis because one of the main assumptions of one-sample t-tests is that all the data are independent/not-correlated. Here the data are clearly correlated since each data-point comes from the same population.

Comment 20 - for each factor (for example, self-harm method), the authors perform a separate statistical test for each of the levels within this factor in tables 2 and 3. For example, for self-harm method, there were 11 tests performed. By chance, 5% of the results may be significant since that is what a p-value of 0.05 represents. Thus, the authors should adjust the p-values for the number of tests performed for each factor.

Other Notes

“The MPMI for 2020 was significantly lower than the MPMI for 2019 based on repeated measured ANOVA analysis (p < 0.001)”. Please include the F-statistic and degrees of freedom when reporting ANOVA results. Also, clarify in the methods whether it is a one, two, or three-way ANOVA since it seems two other factors were considered in figure 1: 2019/2020 and Rural/Urban. The authors imply they performed post-hoc comparisons in MPMI at specific time-points (ex. “ The weekly MPMI value was higher (+2.35%) during January 2020 than during the same period in 2019 but decreased sharply beginning the first week of February (week 6 of 2020) and remained significantly lower than in 2019 until it returned to the previous year’s level in the first week of April.”). Please indicate in the methods how post-hoc tests were conducted and what multiple testing comparison adjustments were made.

7. PLOS authors have the option to publish the peer review history of their article (what does this mean?). If published, this will include your full peer review and any attached files.

Reviewer #1: No

Reviewer #2: No

Reviewer #3: No

---

## [Author Response · Author response to Decision Letter 1]

25 Apr 2023

Here is a point-by-point response to the reviewers’ comments and concerns.

Comments from Reviewer 2

Thank you for the revised article. When reading it again, I feel that the article needs major language quality control. There are missing words in the text and the authors mix up past tense with present tense. Please find some (minor) comments below referring to this.

Please accept my sincere thanks for reviewing our paper. I have responded to each of the comments below, so please read them and review the manuscript.

Comment 1: -The authors still use ”area” and ”region” and "province" in the article. Be consistent in using one of these words.

Response: The three terms have been unified into "region". Your comment is greatly appreciated.

Comment 2: RESULTS

-the first 6 lines in RESULTS belongs to the Methods

Response 2: We moved the contents to the Methods.

Comment 3: line 6: What time period /year are you referring to in this sentence? "A total of 32,778,938 people lived in the urban regions, and 14,736,844 people lived in the eight rural provinces".

Response 3: The numbers were measured at the mid of 2020 (1st July) and we mentioned that in the methods section.

Comment 4: You wrote "eight rural provinces" but you didn't provide information on how many urban regions there were. You can either delete the 8 or write also number of urban areas.

Response 4: In the second paragraph of the section titled "Study design and data collection", we have included information about urban and rural regions. Please review the revised version.

Comment 5: line 9: Please rephrase this sentence. The wording is imprecise: "In the chi-square post-hoc test for age-specific distribution, the proportion in the 15 – 24 and 25 – 34–year–old groups in the rural area, and the proportion in the over–45–year–old group in the urban area increased relative to each other (p < 0.001)."

Furthermore, the young (15-24 and 25-34) are overrepresented in urban area according to the table 1, and not rural area, while those aged 45 and over are overrepresented in rural area, and not urban area.

-Also, you do not have age group 45+ according to the table 1. You can write instead: those aged 45 and over or patients aged 45 and over.

Response 5: Thank you for providing such a detailed response. We have revised the manuscript as much as possible in accordance with your suggestions. If you could review it again, we would greatly appreciate it.

Comment 6: line 11: The authors used the past tense and then the present tense. Use past tense in relation to previous text: "Across both study regions, poisoning was the most common method of self-harm, while traffic accidents are..." (WERE)....

Response 6: Please review the revised version.

Comment 7: You are not comparing traffic accidents with laceration/stabbing. Do you mean AND instead of THAN in the sentence?: "Across both study regions, poisoning was the most common method of self-harm, while traffic accidents are more common in rural regions THAN laceration/stabbing..."r

Response 7: We revised. Please review again.

-Also I suggest that you place p-values inside the sentence: …poisoning was the most common method of self-harm (p < 0.001), while traffic accidents are more common in rural regions (p < 0.001), and laceration/stabbing in urban areas (p = 0.006).

Response 7: A Chi-square test was performed to determine the independence of self-harm methods and study regions, and we presented one p-value for all methods of self-harm. Therefore, we revised the sentence as follows: “It was found in both study regions that poisoning was the most common method of self-harm, and that was overrepresented in rural regions. Laceration/stabbing, hanging/chocking, struck by object and fall were more overrepresented in in urban regions, but submersion, burn and traffic accident were more prevalent in rural regions (p < 0.001).”

Comment 8: line 13: Accoriding to the table 1, the proportion of patients in outpatient unit was higher in rural than urban regions (41 vs 37): "The proportion of patients who visited EDs through outpatient units was high in urban regions (p < 0.001)".

Response 3: Please accept our sincere apologies. We corrected that.

Comment 9: line 14: The word WAS is missing in the sentence. Please rephrase this sentence: A higher proportion of patients presented to EDs in an alert mental state WAS in urban regions than in rural regions from chi-square post-hoc test (p<0.001).

Response 9: We revised sentence as “The proportion of patients in an alert mental state who attended EDs was higher than in urban regions, whereas rural regions had a higher proportion of unresponsive patients.”. 

 To determine the independence of mentality distribution between two regions, a chi-square test is appropriate. Accordingly, the results of the post-hoc chi-square test have been excluded.

Comment 10: site 8, line 2: There are missing some words in this sentence: "….decreased sharply beginning the first week of February…"

Response 3: We revised as “…but decreased sharply at the first week….”

Comment 11: site 9, line 8: Change the word order in this sentence: "The proportion of self-harm patients in two age groups (15 – 24 and 25 – 34) increased significantly in 2020 (p < 0.001 and p = 0.046, respectively), whereas decreased significantly in the 35 – 44, 45 – 54, 55 – 64, 65 – 74, 75 – 84, 85 – 94 and over-95 age groups (p <0.001, p = 0.003, p < 0.001, p = 0.004, p = 0.045, p <0.001 and p = 0.010, respectively)."

Response 11: We revised. Please review again.

Comment 12: -site 9 line 11: Change the word ”and” to "while"in this sentence:

According to the self-harm method only the proportion of laceration/stabbing injuries increased to 29.5%,% (p = 0.006),AND methods such as being struck by an object, hanging/choking, falling, burningfall, submersion, burn, and traffic accidents decreased significantly (p = 0.015, p < 0.001, p = 0.004, p = 0.016, p = 0.004 and p = 0.041 respectively).

Response 12: Thank you. We revised that.

Comment 13: site 11 Table 3: In table 3 the authors calculate annual visit rate after self-harm (VRSH) by age and sex. In the method section, you only describe the weekly VRSH, but do not mention anything about the annual VRSH by age and sex.

Response 13: The related contents have been added in methods section. Please review again.

Comment 14: site 11 line 1-2: The authors wrote p=0.35. Do you mean p=0.356?" Female self-harm visits per 100,000 people increased to 78.4 in 2020, showing a sustained increase over the previous five years (p = 0.011), whereas male self-harm visit rates did not significantly increase over time (p = 0.35) (table 3)."

Response 14: As a result of changes in statistical methods, the results and p-values have been slightly altered. Results have been updated. We request that you review this again.

Comment 15: site 11 line 2: The p-value for group aged 25-34 is <0,001 according to table 3, and not p=0.008. "There was a significant increase in standardized self-harm visit rates for groups aged 15 –24 (p = 0.003) and 25 – 34 (p = 0.008) compared to the previous five years, but a decrease for those 75 – 84 (p = 0.049), 85 – 94 (p = 0.010), 95 years and older (p = 0.011)"

-Also at the end of the sentence “and” is missing before “95 years and older (p = 0.011)”.

Response 15: Thank you. We revised that.

Comment 16: site 12: The first 3 lines belongs to methods.

Response 16: Thank you. We moved the contents and revised.

Comment 17: site 12 line 9: The word WHICH is missing in this sentence: "In rural regions, the lag time between decreases in the MPMI and VRSH were 0.000 (IQR: 0.000 – 0.250) weeks, WHICH is slightly shorter than 0.500 (IQR: 0.000 -1.250) weeks seen in urban regions, but not statistically significant (p = 0.245)."

Response 17: Thank you. We corrected.

Comments from Reviewer 3

Comment 1: the authors did not respond to this comment. The response is simply a copy of one of my other original comments for the introduction. Again, please include what is the motivation for comparing rural vs urban self-harm ED usage in the introduction.

Response 1: We cited studies in East Asia which divided urban and rural regions similarly to our study, compared the effectiveness of physical distance based on regional characteristics, and noted differences in effectiveness among regions based on regional characteristics. We also indicated that the purpose of the study is determining the differences of characteristics of self-harm patients according to region in introduction section. Please review again.

Comment 2: authors report results were corrected for multiple comparisons using the Bonferroni method and this was included in the methods. Upon review, no mention of Bonferroni correction was reported in the methods section. Please also indicate whether p-values in all the tables are corrected for multiple comparisons in the table legends.

Response 2: In response to your comments, we have modified the statistical methodology. In table 1, we conducted a chi-square test without a post-hoc test. Therefore, multiple comparisons are not necessary. Please review again.

Comment 3: In the methods, the authors write: “The Student t-test or Mann-Whitney U test was used to compare the characteristics of the two study regions (Table 1)”. Which tests were t-tests/u-tests in Table. 1? Aren’t the tests in Table. 1 all Chi-square tests of independence?

Response 3: Your point is absolutely correct. We used a unified compare the distribution between two groups (Chi-square tests or Fisher's exact test of independence). Thank you for your comments.

Post-hoc Chi-square tests are only done when contingency tables exceed a 2x2 design. In table 1, they do not need to be done for sex and urban/rural area. Post-hoc Chi-square tests may be done for factors with more than 2-levels, such as the age factor that has 9 levels. Post-hoc tests are typically performed using individual pairs of those levels; ie 9C2 = 36 pairs to test (ex. Ages 15-24 and 25-34 vs. sex, then ages 15-24 and 35-44 vs. sex….etc). Please clarify in the methods and results how the post-hoc Chi-square tests were done. Additionally, corrections for multiple testing will need to be done for the 36 post-hoc chi-square tests for age. Due to the high number of tests, less stringent correction methods such as Bonferroni-Holm can be considered. Here is a resource that provides some guidance on post-hoc Chi-square tests. https://alanarnholt.github.io/PDS-Bookdown2/post-hoc-tests-1.html.

Response: We are sorry for confusion. And thank you very much for your valuable feedback. After serious discussion, the authors found it unreasonable to display all 36 comparisons (9C2 = 36), because the description of the numbers did not contribute much to the study findings. Therefore, I deleted the post-hoc test results. It is possible to confirm the difference of distribution of variable between the two regions by using the Chi-square test.

Comment 4: Additionally, if more than 20% of the cells have < 5 counts for each Chi^2 test, Fisher’s exact test should be used instead of Chi-square test of independence.

Response 4: Your point is very accurate, thank you. We revised the methods section. We reanalyzed the data using Fisher's exact test for "route to ED". However, the p-value was same as < 0.001.

Comment 5: where are the contingency coefficients (that the authors report they calculated) reported in table 1? They are also not mentioned in the results.

Response 5: There was no contingency coefficient in Table 1. And the statistical method was only unified with the Chi-square test, which was used to determine whether the distribution of variables between two regions was independent.

Comment 6: one-sample t-tests require a sample of observations, and then comparing the data to the expected mean. For the results in Table. 2, the authors state they are comparing the proportion (for example, of females) in 2020 to the average of the previous 5 years. The proportion of females in 2020 is a single observation, not a sample of observations, so it is not possible to do a t-test. How did the authors do t-tests for this data?

When comparing proportions, it is more appropriate to use Chi-square tests of independence. If you have an expected proportion (for example, average proportion of females from 2015 - 2019), you can perform a one-sample Chi-square test by comparing your observed proportion (ex. Proportion of females in 2020) to the expected proportion.

Response 6: You made a very valid point. However, our primary objective was to compare the values of the early pandemic era in 2020 with the trend changes of the previous five years.

One sample chi-square test is used to determine whether the distribution of the variable is the same as the expected distribution (https://www.spss-tutorials.com/spss-one-sample-chi-square-test/). Therefore, the one-sample chi-square test has the same drawback as the one-sample t-test.

 Here is an example of a graph illustrating the proportion of female patients who self-harm each year. If the values in the upward trend are analyzed with a one-sample t-test, the 2020 value will appear significantly different from the average of previous years, as shown in the graph below. 

To solve this problem, we consulted to statistician and utilized a new trend analysis method.

As a result, we decided a new statistical method, joinpoint regression is employed to determine if a statistically significant change has taken place between the APC value for 2015-2019 and that of 2019-2020.

 Below are the results of the joinpoint regression for the annual changes in the proportion of males among self-harm patients. From 2015 to 2019, there had been a gradual decline, followed by a sharp decline in 2020. We obtained the APC values of these two intervals and verified if there was a significant difference between the APC values of the two segments (blue versus green).

Through joinpoint analysis, we calculated the AAPC and APC and their confidence intervals. And the results represented in Tables 2 and 3. It would be greatly appreciated if you could review it again.

Comment 7: Table 3 legend indicates “A Shapiro-Wilk test was conducted to test for normality of the variables (2014-2018), and a one-sample t-test was performed if the normality test passed (p > 0.050).” First, the data ranges from 2015-2020, so shouldn’t the authors have conducted Shapiro-Wilk tests on 2015-2019 values? Second, for one-sample t-tests, it seems the authors may have used emergency department rates from 2015-2019 as the “sample” and the emergency department rate in 2020 as the “expected mean” to compare the sample with. If this is the case, then the one-sample t-test is NOT an appropriate statistical test for this analysis because one of the main assumptions of one-sample t-tests is that all the data are independent/not-correlated. Here the data are clearly correlated since each data-point comes from the same population.

Response 7: The phrase "(2014-2018)" was mistyped as "(2015-2019)"

Comment 8: for each factor (for example, self-harm method), the authors perform a separate statistical test for each of the levels within this factor in tables 2 and 3. For example, for self-harm method, there were 11 tests performed. By chance, 5% of the results may be significant since that is what a p-value of 0.05 represents. Thus, the authors should adjust the p-values for the number of tests performed for each factor.

Response 8: According to all of review opinions, a one-sample t-test would not be appropriate to determine an increase or decrease. We re-analyzed the data using a new tool - Joinpoint regression, which is a method for comparing a change between 2019 and 2020 to a trend over the previous five years (please refer to Response 6)- after consulting a statistician. Although there is little change in the results, I believe it would be reasonable to revise the results. It would be greatly appreciated if you could review it again.

Other notes

“The MPMI for 2020 was significantly lower than the MPMI for 2019 based on repeated measured ANOVA analysis (p < 0.001)”. Please include the F-statistic and degrees of freedom when reporting ANOVA results. Also, clarify in the methods whether it is a one, two, or three-way ANOVA since it seems two other factors were considered in figure 1: 2019/2020 and Rural/Urban. The authors imply they performed post-hoc comparisons in MPMI at specific time-points (ex. “ The weekly MPMI value was higher (+2.35%) during January 2020 than during the same period in 2019 but decreased sharply beginning the first week of February (week 6 of 2020) and remained significantly lower than in 2019 until it returned to the previous year’s level in the first week of April.”). Please indicate in the methods how post-hoc tests were conducted and what multiple testing comparison adjustments were made.

Response: 

We completely agree with your comments. A two-way ANOVA was used for the analysis. Therefore, we remarked the F and p-value obtained from the analysis in head of the results section. Additionally, all our comparison targets fell into two groups (urban versus rural, 2019 versus 2020), so an additional post-hoc analysis was not conducted.

We appreciate the advice you provided.

---

## [Editor Report · Decision Letter 2]

16 May 2023

Physical distancing and emergency medical services utilization after self-harm in Korea during the early COVID-19 pandemic: A nationwide quantitative study

PONE-D-22-22495R2

Dear Dr. Hong,

We’re pleased to inform you that your manuscript has been judged scientifically suitable for publication and will be formally accepted for publication once it meets all outstanding technical requirements.

Kind regards,

Pei Boon Ooi, Ph.D.

Academic Editor

PLOS ONE

Additional Editor Comments (optional):

Thank you for your revision and the efforts in addressing all the comments and suggestions.
---

## [Editor Report · Acceptance letter]

19 May 2023

PONE-D-22-22495R2 

Physical distancing and emergency medical services utilization after self-harm in Korea during the early COVID-19 pandemic: A nationwide quantitative study 

Dear Dr. Hong:

I'm pleased to inform you that your manuscript has been deemed suitable for publication in PLOS ONE. Congratulations! Your manuscript is now with our production department. 

Kind regards, 

on behalf of

Dr. Pei Boon Ooi 

Academic Editor

PLOS ONE